# Dynamic Low-Rank Sparse Adaptation for Large Language Models

**Weizhong Huang**[1]    **Yuxin Zhang**[1]    **Xiawu Zheng**[1,3,4]    **Yang Liu**[2]    **Jing Lin**[2]
**Yiwu Yao**[2]    **Rongrong Ji**[1,3*]

[1]Key Laboratory of Multimedia Trusted Perception and Efficient Computing,
   Ministry of Education of China, Xiamen University, 361005, P.R. China.
[2] Huawei Technologies.   [3]Institute of Artificial Intelligence, Xiamen University.
[4] Peng Cheng Laboratory, Shenzhen, China.

## Abstract

Despite the efficacy of network sparsity in alleviating the deployment strain of Large Language Models (LLMs), it endures significant performance degradation. Applying Low-Rank Adaptation (LoRA) to fine-tune the sparse LLMs offers an intuitive approach to counter this predicament, while it holds shortcomings include: 1) The inability to integrate LoRA weights into sparse LLMs post-training, and 2) Insufficient performance recovery at high sparsity ratios. In this paper, we introduce dynamic **Lo**w-rank **S**parse **A**daptation **(LoSA)**, a novel method that seamlessly integrates low-rank adaptation into LLM sparsity within a unified framework, thereby enhancing the performance of sparse LLMs without increasing the inference latency. In particular, LoSA dynamically sparsifies the LoRA outcomes based on the corresponding sparse weights during fine-tuning, thus guaranteeing that the LoRA module can be integrated into the sparse LLMs post-training. Besides, LoSA leverages Representation Mutual Information (RMI) as an indicator to determine the importance of layers, thereby efficiently determining the layer-wise sparsity rates during fine-tuning. Predicated on this, LoSA adjusts the rank of the LoRA module based on the variability in layer-wise reconstruction errors, allocating an appropriate fine-tuning for each layer to reduce the output discrepancies between dense and sparse LLMs. Extensive experiments tell that LoSA can efficiently boost the efficacy of sparse LLMs within a few hours, without introducing any additional inferential burden. For example, LoSA reduced the perplexity of sparse LLaMA-2-7B by 68.73↓ and increased zero-shot accuracy by 16.32%↑, achieving a **2.60×** speedup on CPU and **2.23×** speedup on GPU, requiring only **45 minutes** of fine-tuning on **a single** NVIDIA A100 80GB GPU. Code is available at `https://github.com/wzhuang-xmu/LoSA`.

## 1 Introduction

The development of large language models (LLMs) (Zhang et al., 2022; Touvron et al., 2023a;b) has marked substantial advancements in the field of natural language processing (Achiam et al., 2023). As the scale of these models increases, they demonstrate enhanced capabilities in understanding and generating across diverse contexts (Kaplan et al., 2020; Brown et al., 2020). Nevertheless, the exponential growth in model size presents formidable challenges for deployment and inference, primarily due to escalated computational demands and latency (Zhu et al., 2023). To mitigate these issues, a variety of model compression strategies have been developed. Techniques such as sparsity (Frantar & Alistarh, 2023; Sun et al., 2023; Dong et al., 2024; Ma et al., 2023; Xia et al., 2023; An et al., 2024; Huang et al., 2025a), quantization (Egiazarian et al., 2024; Xiao et al., 2023; Xu et al., 2024b; Lin et al., 2023), and knowledge distillation (Ko et al., 2024; Hsieh et al., 2023; Gu et al., 2023; Agarwal et al., 2024) have proven effective in reducing model size while largely preserving their original efficacy, thus enhancing the feasibility of deploying large models in practical applications.

---

*Corresponding author: rrji@xmu.edu.cn

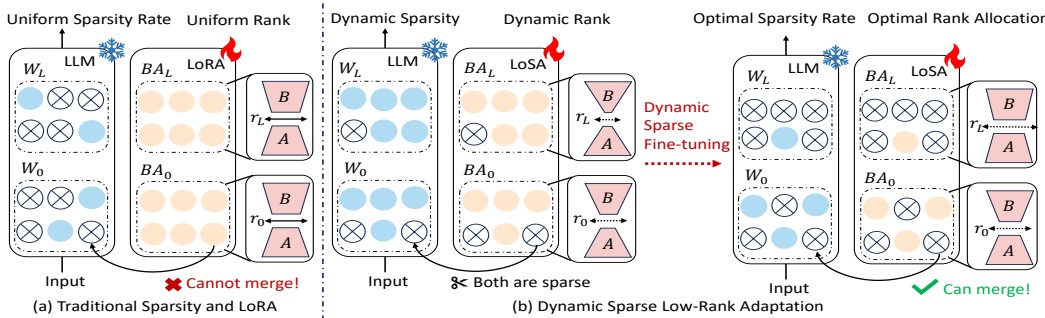

Figure 1: Comparing traditional sparse LLM combined with LoRA to our LoSA method: (a) Traditional LLM sparsity methods employ uniform sparsity rates, and LoRA also uses uniform ranks. Additionally, LoRA weights cannot be merged into the sparse LLM weights. (b) LoSA performs dynamic sparse low-rank adaptation on LLMs, simultaneously applying sparsity to both LLM and low-rank adaptation. Moreover, LoSA dynamically determines the layer-wise sparsity rates based on representation mutual information and allocates the ranks of the low-rank adaptation according to the reconstruction errors of the sparse LLM.

Among the diverse array of model compression techniques, sparsity emerges as a prominent method for diminishing both the size and computational demands of LLMs (Li et al., 2023b; Lu et al., 2024; Frantar & Alistarh, 2023; Sun et al., 2023). Notable implementations such as SparseGPT (Frantar & Alistarh, 2023) and Wanda (Sun et al., 2023) have effectively demonstrated one-shot sparsity, achieving substantial reductions in model size while largely maintaining performance. These strategies also enhance inference speed on CPUs and GPUs through integration with specialized libraries such as DeepSparse (NeuralMagic, 2021) and nm-vllm (NeuralMagic, 2024). However, these approaches encounter performance degradations at high sparsity levels. Consequently, fine-tuning are essential to recuperate the efficacy of sparse LLMs (Guo et al., 2024), thereby ensuring that they remain robust and effective in practical applications.

Leveraging parameter-efficient fine-tuning (PEFT) methods (Houlsby et al., 2019; Lester et al., 2021; Hu et al., 2021) offers a compelling approach to fine-tune sparse LLMs without necessitating the adjustment of the entire model's parameters. By incorporating only a minimal number of additional parameters, PEFT methods circumvent the resource-intensive demands typically associated with full-model fine-tuning. Among various PEFT strategies, Low-rank Adaptation (LoRA) (Hu et al., 2021) distinguishes itself through its innovative use of two low-rank matrices. These matrices are integrated during the fine-tuning phase and subsequently can be merged with the original model weights. This integration effectively preserves the original structure of the model while also eliminating delays during inference, thereby streamlining the deployment process and improving the model's performance after fine-tuning.

However, directly employing the existing LoRA method to fine-tune sparse LLMs faces several critical issues: **1) Incompatibility between sparse LLMs and LoRA.** The weights refined through LoRA cannot be seamlessly integrated into sparse LLMs (Zhang et al., 2023a). Retaining LoRA weight matrices leads to increased inference delays in sparse LLMs (Table 9), thereby compromising the compression and acceleration advantages initially gained from sparsity. **2) Static setup of uniform sparsity rates.** Existing SparseGPT (Frantar & Alistarh, 2023) and Wanda (Sun et al., 2023) methods predefine a uniform sparsity rate for each layer; however, the relative importance of each layer may change during the fine-tuning process. The setting of uniform layer-wise sparsity rates impairs the performance of the sparse model. **3) Static determination of the LoRA matrices' rank.** LoRA's approach of predetermining the rank for each layer fails to account for the variability in layer-wise reconstruction errors that arise during the sparsity of LLMs. Consequently, this results in a uniform allocation of fine-tuning parameters across different layers, which is insufficient for achieving satisfactory fine-tuning performance (Zhang et al., 2023b).

To address the issues mentioned above, we propose dynamic **Lo**w-rank **S**parse **A**daptation (**LoSA**) for LLMs, a method that seamlessly integrates low-rank adaptation into LLM sparsity. LoSA attains this objective through three primary innovations. Firstly, to maintain compatibility between the sparse LLM weights and low-rank adaptation, we dynamically sparsify the low-rank adaptation,

ensuring they align with the sparsity patterns of the LLM weights. Furthermore, we dynamically adjust the layer-wise sparsity rates of the LLM using Representation Mutual Information (RMI) (Tishby et al., 2000; Zheng et al., 2022). Firstly, we derive that RMI can be used as a metric to determine the importance of each layer in LLMs according to the Information Bottleneck (IB) principle (Tishby et al., 2000). Furthermore, we approximate the calculation of RMI using normalized Hilbert-Schmidt Independence Criterion (HSIC) (Gretton et al., 2005; Kornblith et al., 2019). This means that we only need to obtain the feature map of LLM to calculate RMI, allowing us to determine the importance of each layer and efficiently set the layer-wise sparsity rate during fine-tuning. Lastly, we dynamically determine the ranks for the low-rank adaptation based on reconstruction errors, which allows us to allocate a larger fine-tuning parameter budget to layers with greater reconstruction errors. This strategy not only achieves a rational distribution of fine-tuning parameters but also maximizes the reduction of reconstruction errors in sparse LLMs, enhancing overall fine-tuning performance.

Extensive experimental results demonstrate the efficiency and effectiveness of our proposed LoSA for sparsifying representative LLMs, including LLaMA-1 (Touvron et al., 2023a), LLaMA-2 (Touvron et al., 2023b), LLaMA-3 (Meta, 2024a), LLaMA-3.1 (Meta, 2024b), OPT (Zhang et al., 2022) and Vicuna (Chiang et al., 2023) with their parameter sizes ranging from 7 billion to 70 billion. Remarkably, our approach reduces the perplexity of a sparse LLaMA-2-7B (Touvron et al., 2023b) model with sparsity ratio of 70%, obtained using Wanda (Sun et al., 2023) method, by 68.73↓, while achieving an accuracy improvement of 16.32%↑ across seven downstream datasets. Moreover, it achieves a **2.60**× speedup on CPU and **2.23**× speedup on GPU, requiring only **45 minutes** of fine-tuning on a **single** NVIDIA A100 80GB GPU.

## 2 METHODOLOGY

### 2.1 PRELIMINARIES

**Notation.** In this study, we use uppercase letters (*e.g.*, $X, Y$) to denote random variables. Bold typeface represents vectors (*e.g.*, $\boldsymbol{x}, \boldsymbol{y}$), matrices or tensors (*e.g.*, $\boldsymbol{X}, \boldsymbol{Y}$). Calligraphic font indicates loss functions (*e.g.*, $\mathcal{L}$).

**Problem Formulation.** Following the approach introduced by SparseGPT (Frantar & Alistarh, 2023), we conceptualize the implementation of sparsity in LLMs as a layer-wise reconstruction problem. Our objective is to minimize the difference in output between each layer of a sparse LLM and its corresponding dense counterpart. Consider a dense LLM composed of $n$ layers, where the weight matrix of the $i$-th layer is denoted as $\boldsymbol{W}_i \in \mathbb{R}^{c_{out} \times c_{in}}$, $c_{in}$ and $c_{out}$ representing the number of input and output channels, respectively. The input feature maps are represented by $\boldsymbol{X}_i \in \mathbb{R}^{c_{in} \times d}$, where $d$ is the hidden dimension. The sparsity mechanism in an LLM involves applying a binary mask $\boldsymbol{M}_i \in \{0, 1\}^{c_{out} \times c_{in}}$ to the weight matrix $\boldsymbol{W}_i$, which selectively eliminates individual elements.

In this study, we explore the integration of low-rank adaptations with sparsity methods for LLMs. Our approach incorporates low-rank adaptations into the framework of layer-wise reconstruction error optimization. We introduce low-rank adaptation for the $i$-th layer as $\boldsymbol{B}_i \in \mathbb{R}^{c_{out} \times r_i}$ and $\boldsymbol{A}_i \in \mathbb{R}^{r_i \times c_{in}}$, with $r_i$ representing the rank of the adaptation. Simultaneously, we define the ranks of the low-rank adaptations for all $n$ layers collectively as $\boldsymbol{r} = (r_1, r_2, ..., r_n)$, and the sparsity rates for all $n$ layers as $\boldsymbol{s} = (s_1, s_2, ..., s_n)$. Consequently, the low-rank sparse adaptation for LLMs can be viewed as the following optimization problem:

$$\min_{\boldsymbol{M}, \boldsymbol{BA}} \sum_{i=1}^{n} || \underbrace{\boldsymbol{W}_i * \boldsymbol{X}_i - (\boldsymbol{M}_i \odot (\boldsymbol{W}_i + \boldsymbol{B}_i \boldsymbol{A}_i)) * \boldsymbol{X}_i}_{\mathcal{L}_i} ||_2^2,$$

$$s.t. \quad \frac{||\boldsymbol{M}_i||_0}{c_{out} \cdot c_{in}} = s_i, \frac{1}{n} \sum_{i=1}^{n} s_i = \Theta, \frac{1}{n} \sum_{i=1}^{n} r_i = \Omega,$$

(1)

where $*$ denotes matrix multiplication, $\odot$ represents Hadamard product, $|| \cdot ||_2$ signifies the $\ell_2$ norm, $\mathcal{L}_i$ denotes reconstruction error of the $i$-th layer, and $||\boldsymbol{M}_i||_0$ indicates the number of 0 elements in matrix $\boldsymbol{M}_i$.

The proposed formula integrates LLM sparsity and low-rank adaptation into a unified optimization objective, framing it as a constrained layer-wise reconstruction problem. This approach offers advantages over the conventional method of optimizing sparsity and fine-tuning separately. By employing joint optimization, we aim to improve the accuracy of sparse LLM and low-rank adaptation can be merged into sparse LLM.

Our optimization objective underscores the necessity of determining three critical parameters in the joint optimization process for sparse fine-tuning: sparsity mask $M$, layer-wise sparsity rates $s$, and the rank allocations for each layer $r$. To derive the sparsity mask $M$, we leverage existing sparsity methods such as SparseGPT (Frantar & Alistarh, 2023) and Wanda (Sun et al., 2023). Notably, our approach is designed to be compatible with any sparsity method, offering the potential to enhance the accuracy of any sparse LLM. In the subsequent sections, we will elucidate our methodology for determining the layer-wise sparsity rates $s$ and the layer-wise rank allocations $r$, which are crucial components of our optimization framework.

## 2.2 LAYER-WISE SPARSITY RATE

**Motivation.** The current SparseGPT (Frantar & Alistarh, 2023) and Wanda (Sun et al., 2023) methods choose a uniform sparsity rate for each layer to sparsify LLMs, largely because determining layer-wise sparsity rates requires sorting the weight importance of each layer. For LLMs with billions of parameters, this is challenging and time-consuming due to computational bottleneck. However, uniform sparsity across layers is not optimal, as the contributions of each layer to the final performance vary significantly (Yin et al., 2023). Applying the same sparsity rate to all layers risks removing important weights (Lu et al., 2022; Wang et al., 2020). This leads us to consider how to overcome computational bottleneck and quickly determine the layer-wise sparsity rates for LLMs?

In this paper, we propose a metric based on Representation Mutual Information (RMI) (Bachman et al., 2019; Tschannen et al., 2019) for efficiently determining layer-wise sparsity rates. Specifically, the calculation of RMI relies on the feature map of each layer, allowing us to evaluate the importance of a layer by simply extracting its feature maps. This RMI-based approach for determining layer-wise sparsity rates significantly reduces computational complexity. We will provide a detailed explanation of this method below.

**RMI for Sparsity.** The Information Bottleneck (IB) (Tishby et al., 2000) principle elucidates the process of balancing mutual information between input and output representations in LLMs. For the hidden representation of the $i$-th layer $X_i$, the goal is to minimize the mutual information $\mathrm{I}(X; X_i)$ between input $X$ and $X_i$ to reduce inter-layer redundancy, while simultaneously maximizing the mutual information $\mathrm{I}(X_i; Y)$ between $X_i$ and output $Y$ to ensure the layer retains task-relevant information for accurate predictions of $Y$. Specifically, it can be formulated as:

$$\min \mathrm{I}(X; X_i) - \beta \mathrm{I}(X_i; Y), \tag{2}$$

where $\beta$ is a trade-off parameter that balances information compression and the retention of task-relevant information. Given a LLM consist of $n$ layers, we aim to minimize redundancy not only between the input and individual layers but also across different layers within the model. To generalize the IB principle in this multi-layer context, we extend the objective as follows:

$$\min \sum_{i=1}^{n} \sum_{j=i+1}^{n} \left( \mathrm{I}(X; X_i) + \mathrm{I}(X_j; X_i) \right) - \beta \mathrm{I}(X_i; Y). \tag{3}$$

In this expanded formulation, the term $\mathrm{I}(X_j; X_i)(i \neq j, i, j = 1, \ldots, n)$ represents the mutual information between two distinct layers, capturing the redundancy between them. The objective is to minimize this inter-layer mutual information so that the representations learned by different layers are as independent as possible. This implies that layers that are highly correlated with others are less important. Therefore, the RMI between different layers $\mathrm{I}(X_j; X_i)$ can serve as an accurate and robust indicator of the importance of LLM layers (Zheng et al., 2021).

**Algorithm Design.** Now we describe how to obtain the layer-wise sparsity rate of LLM using RMI. We use $X_1, \ldots, X_n$ to represent the hidden representations in each layer, the RMI between the

$i$-th layer and the $j$-th layer is denoted as $\mathrm{I}(X_i, X_j)$. As mentioned above, a layer correlated to other layers is less important. Therefore, the importance of a specific layer $i$ is defined as:

$$p_i = e^{-\sum_{j=1, j \neq i}^{n} \mathrm{I}(X_i, X_j)}, \tag{4}$$

where $e$ is natural constant. With the layer-wise importance $\boldsymbol{p} \in \mathbb{R}^{n \times 1}$, determining the layer-wise sparsity rate is transformed into a linear programming problem. Formally, the layer-wise sparsity rate $\boldsymbol{s} \in \mathbb{R}^{n \times 1}$ can be determined as follows:

$$\min_{\boldsymbol{s}} \boldsymbol{p}^T \boldsymbol{s} \; s.t. \; \frac{1}{n} \sum_{i=1}^{n} s_i = \Theta. \tag{5}$$

It should be noted that the RMI mentioned above is challenging to compute in practice, since the distribution in $I(X_i, X_j)$ is intractable and is time-consuming to estimate. Here, we introduce the normalized Hilbert-Schmidt Independence Criterion (HSIC) (Gretton et al., 2005; Zheng et al., 2021; Kornblith et al., 2019) [1] to address this issue. First, we obtain the feature maps of each layer in LLM, denoted as $\boldsymbol{X}_1, \boldsymbol{X}_2, \ldots, \boldsymbol{X}_n$. Therefore, the RMI is calculated as:

$$\mathrm{I}(X_i, X_j) \approx \mathrm{nHSIC}_{\mathrm{linear}}(X_i, X_j) = \frac{\|\boldsymbol{X}_j^T \boldsymbol{X}_i\|_F^2}{\|\boldsymbol{X}_i^T \boldsymbol{X}_i\|_F \|\boldsymbol{X}_j^T \boldsymbol{X}_j\|_F}, \tag{6}$$

where $\| \cdot \|_F$ denotes the Frobenius norm. This means that in Eq. 6, we use the feature maps to estimate the RMI indicator, making the practical calculation of RMI possible. Combining Eq. 5 and Eq. 6, we can determine the importance of each layer of LLM across layers, thereby quickly determining the layer-wise sparsity rate of LLM. The end-to-end time to compute the layer-wise sparsity of LLaMA-2-7B (Touvron et al., 2023b) is only 48 seconds using one NVIDIA A100 80GB GPU, which is quite fast. We demonstrate the effectiveness of the above method in the ablation experiments in Section 3.5.

## 2.3 SPARSITY-AWARE RANK ALLOCATION

**Problem Setup.** Using low-rank adaptation can effectively restore the performance of sparse LLMs. However, once a LLM with non-uniform layer-wise sparsity is obtained, low-rank adaptation fine-tuning faces a critical challenge: how to allocate the layer-wise rank of low-rank adaptation for the non-uniform sparse LLM within a limited fine-tuning parameter budget? LoRA (Hu et al., 2021) uniformly assigns the same rank to all low-rank adaptations, which is inefficient because it fails to account for the variability in layer-wise reconstruction errors across the sparse LLM during fine-tuning (Frantar & Alistarh, 2023; Xu et al., 2024a). Intuitively, layers with higher reconstruction errors should be allocated a larger fine-tuning budget, as this would help reduce the reconstruction errors more effectively. Thus, we propose a sparsity-aware rank allocation algorithm that efficiently distributes the fine-tuning parameter budget across each layer, guided by the layer-wise reconstruction errors of the sparse LLM. The objective is to maximize the overall reduction in reconstruction error during the fine-tuning of the sparse LLM. Details of this algorithm are discussed below.

**Algorithm Design.** The notation $(\mathcal{L}_1, \mathcal{L}_2, \ldots, \mathcal{L}_n)$ denotes the reconstruction errors for $n$ layers, and the average value is $\mathcal{L}_{\mathrm{avg}} = \frac{1}{n} \sum_{i=1}^{n} \mathcal{L}_i$. Meanwhile, $\Omega$ indicates the average rank of all $n$ layers. Consequently, the rank of the $i$-th layer is computed as:

$$r_i = \lfloor \frac{\mathcal{L}_i}{\mathcal{L}_{\mathrm{avg}}} \times \Omega \rceil \tag{7}$$

where $\lfloor x \rceil$ rounds $x$ to the nearest integer. This formula ensures a rational allocation of the fine-tuning budget to each layer based on its reconstruction error. Layers with higher reconstruction errors are assigned larger ranks, while those with lower errors receive smaller ranks, maximizing the reduction in reconstruction error for sparse LLMs. In Section 3.5, we demonstrate the effectiveness of this sparsity-aware rank allocation method through ablation experiments. Additionally, we compare our approach with a strategy that allocates ranks based on sparsity rates, demonstrating that our method achieves better performance.

---

[1] Normalized HSIC is also known as CKA (Kornblith et al., 2019), RV coefficient (Robert & Escoufier, 1976), and Tucker's congruence coefficient (Lorenzo-Seva & Ten Berge, 2006).

For training stability, when low-rank adaptations $\boldsymbol{B}$, $\boldsymbol{A}$ increase their rank, we concatenate random Gaussian initialized parameters $\mathcal{N}(0, \sigma^2)$ to $\boldsymbol{A}$ and zeros to $\boldsymbol{B}$. The above initialization operation is the same as LoRA (Hu et al., 2021), so the layer's output remains unchanged before and after new parameters added. When $\boldsymbol{B}$, $\boldsymbol{A}$ decrease their rank, the new adaptations directly inherit parameters from the original adaptations and the extra parameters are discarded.

## 2.4 DYNAMIC SPARSITY AND ADAPTATION

To achieve better sparse fine-tuning LLMs, we further extend our algorithm to implement a dynamic sparsity and fine-tuning approach. This involves progressively sparsifying an increasing number of weights while simultaneously conducting low-rank adaptation fine-tuning. Dynamic sparsity and fine-tuning ensure the maximum integration of LLM sparsity and low-rank adaptation fine-tuning. We perform $T$ steps of sparsity and fine-tuning, and determine the progressive sparsity rate using the cubic sparsity schedule proposed by (Zhu & Gupta, 2017), as described below:

$$\Theta^t = \Theta^f - \Theta^f \left(1 - \frac{t}{T}\right)^3, \quad t = 1, 2, ..., T \tag{8}$$

where $\Theta^f$ is final sparsity rate and $\Theta^t$ denotes the average sparsity rate of the $n$ layers at step $t$. Furthermore, since the reconstruction error tends to increase with the rising sparsity rate, we linearly increase the average rank $\Omega^t$ at each step, *i.e.*,

$$\Omega^{t+1} = \Omega^t + 1, \quad t = 1, 2, ..., T \tag{9}$$

After calculating the average sparsity rate $\Theta^t$ at step $t$, we first establish the layer-wise sparsity rates $\boldsymbol{s}^t$ using the method outlined in Section 2.2. Subsequently, we simultaneously sparsify the weights of LLM and low-rank adaptation by applying the sparse mask $\boldsymbol{M}^t$, which is derived using either the SparseGPT (Frantar & Alistarh, 2023) or Wanda (Sun et al., 2023) method. This coordinated approach ensures compatibility between the LLM weights and the low-rank adaptations, facilitating the integration of low-rank adaptations into the sparse weights of the LLM after fine-tuning. Once we have established sparse LLM, we then ascertain the layer-wise rank $\boldsymbol{r}^t$ for the low-rank adaptations, employing the rank allocation method described in Section 2.3. The full details of the algorithm are outlined in Algorithm 1.

---

**Algorithm 1:** Dynamic **Lo**w-rank **S**parse **A**daptation (**LoSA**)

---

**Input:** Dense weight of LLM $\boldsymbol{W}$, low-rank adaptation weight $\boldsymbol{BA}$, dynamic steps $T$, target
       sparsity rate $\Theta^f$, initial average rank $\Omega^1$.
**Output:** Sparse fine-tuning LLM.

**for** $t = 1, \cdots, T$ **do**
    Calculate the progressive sparsity rate $\Theta^t$ using Eq. 8;
    Obtain RMI between two layers using Eq. 6;
    Calculate the layer-wise importance $\boldsymbol{p}^t$ by Eq. 4;
    Obtain the layer-wise sparsity rate $\boldsymbol{s}^t$ by Eq. 5;
    Get sparse mask $\boldsymbol{M}^t$ of $\boldsymbol{W^t} + \boldsymbol{B^t A^t}$ through SparseGPT or Wanda;
    Calculate the current average rank by Eq. 9;
    Allocate layer-wise rank $\boldsymbol{r}^t$ by Eq. 7;
    Update low-rank adaptation weight $\boldsymbol{B^t A^t}$;
**end**
Sparse low-rank adaptation is merged into sparse LLM weight to obtain the final sparse LLM.

---

## 3 EXPERIMENTS

### 3.1 EXPERIMENTAL SETTINGS

**Models and Baselines.** We have applied our method to several LLMs, including LLaMA-1 (Touvron et al., 2023a), LLaMA-2 (Touvron et al., 2023b), LLaMA-3 (Meta, 2024a), LLaMA-3.1 (Meta, 2024b), Vicuna (Chiang et al., 2023), and OPT (Zhang et al., 2022), with parameter sizes ranging

Table 1: Perplexity of LoSA for sparse LLMs on WikiText-2 dataset at 50/60/70% sparsity.

| Sparsity | Method | LLaMA-1 | | | LLaMA-2 | | | LLaMA-3 | LLaMA-3.1 | Vicuna |
|---|---|---|---|---|---|---|---|---|---|---|
| | | 7B | 13B | 30B | 7B | 13B | 70B | 8B | 8B | 13B |
| 0% | Dense | 5.68 | 5.09 | 4.77 | 5.12 | 4.57 | 3.12 | 6.05 | 6.18 | 5.94 |
| 50% | SparseGPT | 7.22 | 6.21 | 5.31 | 6.51 | 5.63 | 3.98 | 9.30 | 9.18 | 7.73 |
| | w. LoRA | 6.91 | 6.04 | 5.16 | 6.31 | 5.49 | 3.91 | 8.50 | 8.49 | 6.51 |
| | w. **LoSA** | **6.86** | **5.98** | **5.13** | **6.27** | **5.44** | **3.88** | **8.38** | **8.36** | **6.46** |
| | Wanda | 7.26 | 6.15 | 5.24 | 6.42 | 5.56 | 3.98 | 9.59 | 9.53 | 7.29 |
| | w. LoRA | 6.84 | 6.04 | 5.17 | 6.33 | 5.46 | 3.94 | 8.56 | 8.53 | 6.53 |
| | w. **LoSA** | **6.82** | **5.94** | **5.13** | **6.24** | **5.41** | **3.93** | **8.41** | **8.42** | **6.44** |
| 60% | SparseGPT | 10.41 | 8.43 | 6.81 | 10.14 | 7.88 | 5.10 | 14.85 | 15.10 | 10.02 |
| | w. LoRA | 8.29 | 6.94 | 6.18 | 7.98 | 6.75 | 4.90 | 10.77 | 10.73 | 7.87 |
| | w. **LoSA** | **8.14** | **6.81** | **6.10** | **7.82** | **6.65** | **4.88** | **10.58** | **10.44** | **7.54** |
| | Wanda | 10.69 | 8.75 | 6.56 | 10.79 | 8.40 | 5.25 | 20.02 | 21.51 | 9.54 |
| | w. LoRA | 8.38 | 6.95 | 5.99 | 8.07 | 6.78 | 5.01 | 11.29 | 11.09 | 7.82 |
| | w. **LoSA** | **8.20** | **6.75** | **5.92** | **7.88** | **6.62** | **4.95** | **10.85** | **10.59** | **7.59** |
| 70% | SparseGPT | 26.30 | 19.24 | 12.56 | 27.42 | 20.57 | 9.46 | 40.53 | 39.76 | 21.95 |
| | w. LoRA | 11.48 | 8.95 | 7.54 | 11.06 | 8.99 | 6.32 | 16.50 | 16.05 | 10.19 |
| | w. **LoSA** | **11.20** | **8.71** | **7.21** | **10.82** | **8.82** | **6.13** | **15.74** | **15.41** | **9.92** |
| | Wanda | 85.77 | 55.90 | 17.37 | 79.67 | 48.07 | 11.10 | 112.10 | 109.99 | 44.89 |
| | w. LoRA | 13.46 | 9.90 | 8.34 | 12.57 | 9.65 | 6.50 | 20.25 | 18.98 | 10.42 |
| | w. **LoSA** | **11.75** | **8.79** | **7.98** | **10.94** | **8.86** | **6.30** | **16.59** | **16.46** | **9.94** |

from 7 billion to 70 billion. To further validate the effectiveness of our approach in enhancing the accuracy of existing sparse methods, we selected Wanda (Sun et al., 2023) and SparseGPT (Frantar & Alistarh, 2023) as baselines, and also compare with LoRA (Hu et al., 2021). We set the rank of LoRA to 8 and the remaining training settings for LoRA are the same as those for LoSA.

**Evaluation.** We report perplexity of sparse LLM on WikiText-2 (Merity et al., 2016) dataset and use lm-eval-harness (Gao et al., 2021) to evaluate the zero-shot accuracy on downstream datasets, including HellaSwag (Zellers et al., 2019), Winogrande (Sakaguchi et al., 2021), BoolQ (Clark et al., 2019), OpenBookQA (Mihaylov et al., 2018), PIQA (Bisk et al., 2020), ARC-Easy, and ARC-Challenge (Clark et al., 2018).

**Datasets and Training Details.** We randomly sampled a 10K subset from the Alpaca-GPT4 (Peng et al., 2023) to construct our fine-tuning dataset. We utilized the same calibration dataset as SparseGPT (Frantar & Alistarh, 2023) and Wanda (Sun et al., 2023), which consists of 128 sequences sampled from the C4 training set (Raffel et al., 2020) for sparsification. During the fine-tuning process, we employed the Paged AdamW optimizer (Dettmers et al., 2024), setting a maximum gradient norm of 0.3. The learning rate followed a linear learning rate schedule and set the learning rate to be $2 \times 10^{-4}$. All experiments were conducted on NVIDIA A100 80GB GPUs. We use one GPU for the 7B, 13B, and 8B models, two GPUs for the 30B models, and three GPUs for the 70B models. We set the fine-tuning steps $T = 5$ and initial average rank $\Omega^1 = 6$.

## 3.2 LANGUAGE MODELING

The perplexity results of fine-tuning the sparse LLMs at 50-70% sparsity rate on the WikiText-2 dataset are presented in Table 1. LoSA improves the performance of both the SparseGPT and Wanda methods across models of various parameter sizes and architectures. For instance, when fine-tuning 70 % sparse LLaMA-2-7B with LoSA, the perplexity of the sparse models obtained by SparseGPT and Wanda is reduced by 16.60 and 68.73, respectively. Additionally, our method significantly outperforms LoRA. These results highlight the effectiveness of our method in enhancing the language modeling capabilities of sparse LLMs.

Table 2: Mean Zero-shot accuracy results of LoSA for sparse LLMs on the HellaSwag, Winogrande, BoolQ, OpenBookQA, PIQA, ARC-Easy and ARC-Challenge datasets at 50/60/70% sparsity. The detailed accuracy of each dataset can be found in Appendix J.

| | | LLaMA-1 | | | LLaMA-2 | | | LLaMA-3 | LLaMA-3.1 | Vicuna |
|---|---|---|---|---|---|---|---|---|---|---|
| Sparsity | Method | 7B | 13B | 30B | 7B | 13B | 70B | 8B | 8B | 13B |
| 0% | Dense | 61.74 | 63.84 | 67.41 | 61.88 | 65.00 | 69.14 | 65.62 | 65.93 | 65.53 |
| 50% | SparseGPT | 57.96 | 60.93 | 65.34 | 59.29 | 62.53 | 68.93 | 60.99 | 61.22 | 63.40 |
| | w. LoRA | 59.83 | 62.68 | 66.31 | 60.47 | 63.90 | 69.56 | 63.20 | 64.25 | 63.56 |
| | w. **LoSA** | **60.54** | **63.09** | **66.86** | **61.32** | **64.23** | **69.82** | **64.20** | **64.59** | **64.28** |
| | Wanda | 57.08 | 61.39 | 65.59 | 59.46 | 62.88 | 68.21 | 59.65 | 59.58 | 63.74 |
| | w. LoRA | 59.46 | 63.07 | 66.37 | 60.51 | 63.84 | 69.04 | 62.65 | 63.05 | 64.16 |
| | w. **LoSA** | **60.09** | **63.49** | **67.19** | **60.85** | **64.32** | **69.65** | **63.20** | **63.63** | **64.41** |
| 60% | SparseGPT | 52.72 | 56.57 | 61.63 | 53.90 | 58.20 | 66.27 | 54.05 | 55.80 | 60.25 |
| | w. LoRA | 55.92 | 60.00 | 64.90 | 57.50 | 61.08 | 68.32 | 58.98 | 60.19 | 61.26 |
| | w. **LoSA** | **57.38** | **61.06** | **65.97** | **58.52** | **61.67** | **69.04** | **60.30** | **60.74** | **61.67** |
| | Wanda | 51.98 | 56.19 | 62.46 | 52.51 | 58.19 | 66.27 | 48.90 | 49.78 | 60.19 |
| | w. LoRA | 55.39 | 59.67 | 64.50 | 56.83 | 60.91 | 68.24 | 57.09 | 57.61 | 60.89 |
| | w. **LoSA** | **56.21** | **60.88** | **65.86** | **58.06** | **61.82** | **69.08** | **58.60** | **58.64** | **61.42** |
| 70% | SparseGPT | 43.60 | 48.00 | 53.64 | 43.07 | 47.38 | 60.84 | 43.02 | 42.83 | 48.53 |
| | w. LoRA | 50.41 | 55.00 | 61.63 | 50.76 | 55.16 | 65.72 | 50.36 | 52.33 | 55.15 |
| | w. **LoSA** | **52.74** | **56.53** | **62.37** | **52.51** | **57.16** | **66.41** | **51.99** | **54.06** | **56.40** |
| | Wanda | 37.45 | 40.79 | 53.35 | 35.33 | 38.88 | 58.48 | 35.42 | 36.10 | 42.06 |
| | w. LoRA | 48.30 | 51.83 | 59.06 | 48.18 | 50.62 | 65.27 | 47.02 | 47.52 | 52.34 |
| | w. **LoSA** | **51.20** | **55.31** | **61.57** | **51.65** | **53.00** | **66.12** | **50.37** | **50.33** | **56.37** |

## 3.3 ZERO-SHOT TASKS

We report the improvements in zero-shot accuracy on seven downstream tasks achieved by LoSA for sparse LLMs with 50-70% sparsity, obtained using SparseGPT and Wanda methods in Table 2. Our LoSA method significantly enhances the zero-shot accuracy across different models ranging from 7 billion to 70 billion parameters. Notably, our LoSA method increases the average zero-shot accuracy of the 70% sparse LLaMA-2-7B obtained via Wanda by 16.32%, surpassing LoRA by 3.47%. These experimental results highlight the substantial enhancements in understanding and reasoning capabilities of sparse LLMs brought about by our LoSA approach.

## 3.4 N:M SPARSITY

We extend LoSA to N:M sparsity and adopt a mixed N:8 sparsity (N refers to non-zero weights) configuration following DominoSearch (Sun et al., 2021). We allow different layers to have distinct N values while maintaining a constant overall sparsity ratio. We assign lower N values to more important layers and the N value for each layer are determined using the method described in Section 2.2. The results are presented in Table 3. It is evident that LoSA improves the accuracy of the LLaMA-2-7B under N:M sparsity and outperforms LoRA.

Table 3: Perplexity and mean zero-shot accuracy of mixed N:M sparsity.

| Method | N:M Sparsity | Perplexity | Accuracy |
|---|---|---|---|
| SparseGPT | 2:8 | 103.76 | 33.27 |
| w. LoRA | 2:8 | 22.47 | 41.46 |
| **w. LoSA** | **Mixed 2:8** | **19.97** | **43.77** |
| Wanda | 2:8 | 3006.24 | 32.71 |
| w. LoRA | 2:8 | 56.14 | 38.72 |
| **w. LoSA** | **Mixed 2:8** | **25.41** | **41.91** |

Table 4: The perplexity of the LLaMA-2-7B at different sparsity rates.

| Sparsity | 40% | 50% | 60% | 70% | 80% | 90% |
|---|---|---|---|---|---|---|
| SprseGPT | 6.12 | 6.51 | 10.14 | 27.42 | 115.50 | 1439.35 |
| w. LoRA | 6.08 | 6.30 | 7.98 | 11.06 | 26.35 | 93.16 |
| **w. LoSA** | **6.05** | **6.25** | **7.82** | **10.82** | **21.54** | **84.39** |
| Wanda | 6.07 | 6.42 | 10.79 | 79.67 | 1980.85 | 17151.30 |
| w. LoRA | 6.04 | 6.31 | 8.07 | 12.57 | 36.43 | 335.43 |
| **w. LoSA** | **6.02** | **6.21** | **7.88** | **10.94** | **24.38** | **168.71** |

## 3.5 ABLATION STUDY

**Robustness across Different Sparsity Rates.**   Table 4 presents the results of perplexity for sparse LLaMA-2-7B across different sparsity rate, ranging from 40% to 90%. These results demonstrate that LoSA consistently reduces the perplexity of both SparseGPT and Wanda across all sparsity levels and consistently outperforms LoRA. This validates the robustness and effectiveness of LoSA method at various sparsity rates, ensuring reliable performance even as the pruning level varies.

**Effectiveness of the Proposed Strategies.**   In this paper, we propose three strategies: Layer-wise Sparsity Rate (**LSR**, Section 2.2), Sparsity-Aware Rank Allocation (**SRA, Section 2.3**), and Dynamic Sparsity and Adaptation (**DSA**, Section 2.4). To demonstrate the effectiveness of three strategies, we conduct an ablation study in Table 5. The first row of the table presents the result of the 70% sparse LLaMA-2-7B obtained by Wanda and further fine-tuned by LoSA. We then progressively remove each strategy from LoSA (rows 2-4), combinations of two strategies (rows 5-6), and all three strategies (row 7). We can see that removing any strategy causes a decrease in the final accuracy. The severity of the accuracy drop follows the order: removing three strategies > removing two strategies > removing one strategy. Additionally, among the three strategies, we found that DSA contributed the most to the final accuracy. The experimental results demonstrate that all three proposed strategies contribute to the final accuracy, and using all three strategies achieves the best results.

Table 5: Ablation of the proposed strategies.

| Method | Perplexity | Accuracy |
|---|---|---|
| **LoSA** | **10.94** | **51.65** |
| w/o LSR | 11.36 (+0.42) | 50.62 (-1.03) |
| w/o SRA | 11.44 (+0.50) | 50.94 (-0.71) |
| w/o DSA | 11.78 (+0.84) | 49.90 (-1.75) |
| w/o LSR & SRA | 11.94 (+1.00) | 48.81 (-2.84) |
| w/o LSR & DSA | 12.23 (+1.29) | 48.30 (-3.35) |
| w/o SRA & DSA | 12.03 (+1.09) | 48.54 (-3.11) |
| w/o LSR & SRA & DSA | 12.74 (+1.80) | 47.76 (-3.89) |

Table 6: Experimental results of OPT-13B.

| Method | Sparsity | Perplexity | Accuracy |
|---|---|---|---|
| Dense | 0% | 10.13 | 55.22 |
| SparseGPT | 70% | 20.26 | 47.68 |
| w. LoRA | 70% | 17.73 | 51.34 |
| **w. LoSA** | **70%** | **17.05** | **52.31** |
| Wanda | 70% | 73.70 | 41.51 |
| w. LoRA | 70% | 20.54 | 49.16 |
| **w. LoSA** | **70%** | **19.75** | **50.13** |

**Experimental results for OPT model.**   We show experimental results of LoSA fine-tuning 70% sparse OPT-13B (Zhang et al., 2022) in Table 6. LoSA effectively restores the accuracy of sparse model that are not based on the LLaMA architecture, and its performance is better than LoRA.

**Reconstruction Error vs. Sparsity Rate.**   In Section 2.3, we allocate fine-tuning parameters per layer based on reconstruction error. Table 7 compares this with an alternative method that assigns higher ranks to layers with higher sparsity. Specifically, we used the Wanda to obtain a 70% sparse LLaMA-2-7B model, and carried out LoSA fine-tuning. The results show that reconstruction error-based allocation outperforms sparsity-based allocation, likely because the optimization goal is to minimize error between the dense and sparse LLM, sparsity rate does not adequately reflect changes in reconstruction error. Addtionally, computing the reconstruction error takes only 46 seconds on an NVIDIA A100 80GB GPU, making it a better proxy for rank allocation.

Table 7: Different ways to allocate rank. SR: Sparsity Rate RE: Reconstruction Error.

| Method | Perplexity | Accuracy |
|---|---|---|
| SR | 11.37 | 50.97 |
| **RE** | **10.94** | **51.65** |

**Dynamic Steps.**   We present the ablation study of the dynamic steps T in Figure 2. Steps T determines the frequency at which the sparsity rate increases. A larger T means the sparsity rate increases more slowly, and fewer parameters are removed each time. We show the impact of different values of T on the final perplexity while keeping the number of fine-tuning samples constant. Specifically, we demonstrate the results of fine-tuning a 70% sparse LLaMA-2-7B model obtained using the Wanda method with LoSA. Increasing T appropriately can effectively reduce the model's perplexity. However, a larger T may result in insufficient training of the model after each sparsification, which in turn leads to a further increase in perplexity.

**Rank Budget.** We show the impact of different rank budgets on the LLM's perplexity in Figure 2, with $\Omega^1 = 2, 6, 10, 16$, for the 70 % sparse LLaMA-2-7B obtained using Wanda. All experiments are conducted with a fixed set of 10K fine-tuning samples. Increasing the rank budget appropriately can effectively reduce perplexity, leading to a better recovery of the sparse model's performance. However, since the fine-tuning samples are fixed, further increasing the rank budget results in insufficient training of low-rank adaptation, which causes an increase in perplexity.

## 3.6 ANALYSIS

**Fine-tuning Efficiency.** In Table 8, we demonstrate the fine-tuning efficiency of LoSA. We compared the fine-tuning parameters, time, and GPU memory usage between LoSA and LoRA. LoSA requires fewer fine-tuning parameters, only $1 - s\%$ of LoRA's (where $1 - s\%$ is the sparsity rate), and its GPU memory usage is similar to LoRA. However, since LoSA performs $T = 5$ rounds of sparsification and calculates layer-wise sparsity rates and rank allocation, it takes more time for fine-tuning. Nevertheless, LoSA provides better accuracy and lower inference latency compared to LoRA, and it only requires about an hour of fine-tuning, which we believe is a worthwhile trade-off.

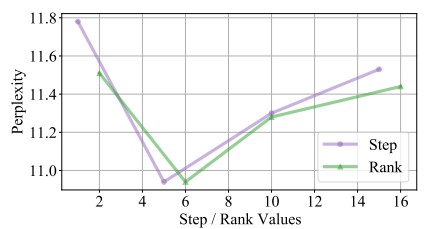

Figure 2: Effect of different steps and ranks on perplexity.

Table 8: Fine-tuning efficiency of LoSA.

| Method | Fine-tuning Params (M) | Fine-tuning Time (min) | Fine-tuning Memory (GB) |
|---|---|---|---|
| Wanda | 0 | 0 | 0 |
| w. LoRA | 20.28 | 13.78 | 53.59 |
| w. LoSA | $20.28 \times (1 - s\%)$ | 45.34 | 53.92 |
| SparseGPT | 0 | 0 | 0 |
| w. LoRA | 20.28 | 21.40 | 53.59 |
| w. LoSA | $20.28 \times (1 - s\%)$ | 73.91 | 53.92 |

**Inference Speedup.** We analyzed the acceleration effect of the sparse LLaMA-2-7B, as shown in Table 9. We measured the end-to-end time of the model generate tokens using the DeepSparse (NeuralMagic, 2021) inference engine on an Intel(R) Xeon(R) Silver 4314 CPU and the nm-vllm (NeuralMagic, 2024) inference engine on a NVIDIA RTX 4090 24GB GPU. Compared to the dense model, our method achieves a remarkable 1.77-2.60× speedup on CPU and 1.71-2.23× speedup on GPU at 50-70% sparsity. In contrast, LoRA cannot merge weights into sparse weights which introduces additional inference latency that increases with higher sparsity rates. This demonstrates the advantage of our LoSA method in maintaining the acceleration performance of sparse LLMs.

Table 9: The end-to-end inference speedup of sparse LLaMA-2-7B on CPU and GPU.

| Device | Sparsity | Dense | 50% | | 60% | | 70% | |
|---|---|---|---|---|---|---|---|---|
| | | | LoRA | LoSA | LoRA | LoSA | LoRA | LoSA |
| CPU | Throughput (tokens/s) ↑ | 3.43 | 5.68 | **6.08** | 6.64 | **7.41** | 7.88 | **8.93** |
| | **Speedup** ↑ | 1.00× | 1.65× | **1.77×** | 1.94× | **2.16×** | 2.29× | **2.60×** |
| GPU | Throughput (tokens/s) ↑ | 57.35 | 79.63 | **97.88** | 88.58 | **111.82** | 98.10 | **127.69** |
| | **Speedup** ↑ | 1.00× | 1.39× | **1.71×** | 1.54× | **1.95×** | 1.71× | **2.23×** |

## 4 CONCLUSION

In this paper, we propose a novel dynamic low-rank sparse adaptation method for the efficient fine-tuning of sparse LLMs. Our method simultaneously sparsifies both LLM and low-rank adaptation, ensuring that low-rank adaptation can be merged into LLM weight post-training, thereby not increasing inference latency. Moreover, we introduce representation mutual information as an effective and efficient metric for dynamically determining the layer-wise sparsity rates during fine-tuning. Additionally, we dynamically adjust the rank of low-rank adaptation based on the layer-wise reconstruction error changes during sparsity, ensuring a efficient fine-tuning budget allocation for each layer. Extensive experiments demonstrate the effectiveness of our method in fine-tuning sparse LLMs.

## REPRODUCIBILITY STATEMENTS

The experimental setup is described in Section 3.1 and code is available at `https://github.com/wzhuang-xmu/LoSA`.

## ACKNOWLEDGMENTS

This work was supported by the National Science Fund for Distinguished Young Scholars (No.62025603), the National Natural Science Foundation of China (No. U21B2037, No. U22B2051, No. U23A20383, No. 62176222, No. 62176223, No. 62176226, No. 62072386, No. 62072387, No. 62072389, No. 62002305 and No. 62272401), and the Natural Science Foundation of Fujian Province of China (No. 2021J06003, No.2022J06001).

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

# A   LIMITATION

Although our LoSA method effectively enhances the accuracy of existing sparsity techniques under different sparsity settings, there is still a gap to achieving lossless high-ratio sparsity for LLMs. This underscores the need for future exploration of more efficient fine-tuning methods to further improve the accuracy of sparse LLMs.

# B   RELATED WORK

**LLM Sparsity.**   Existing sparsity methods (Huang et al., 2025b), including SparseGPT (Frantar & Alistarh, 2023) and Wanda (Sun et al., 2023), enable training-free sparsity of LLMs, effectively eliminating non-essential weights while striving to preserve model performance as much as possible. However, existing sparsity methods can lead to significant accuracy degradation under high

sparsity rates, partly because these methods pre-set uniform layer-wise sparsity rates, overlooking the fact that redundancy levels vary between different layers of LLMs (Song et al., 2024; Men et al., 2024; Chen et al., 2024). OWL (Yin et al., 2023) has recognized this issue and employs heuristic metrics to set the sparsity rates of LLMs inversely proportional to the ratio of observed activation outliers within each layer, thereby achieving a non-uniformly sparse LLMs. While setting non-uniform pruning rates can partially improve the accuracy of sparse LLMs, the accuracy still remains unsatisfactory. Therefore, fine-tuning sparse LLMs to restore their accuracy is necessary. This paper proposes using low-rank sparse adaptation to restore the accuracy of sparse LLMs and dynamically determine the layer-wise sparsity rates using representation mutual information during the fine-tuning process. Representation mutual information (Bachman et al., 2019; Tschannen et al., 2019) have been successfully applied to prune small models such as CNNs and BERT (Zheng et al., 2021; Fan et al., 2021; Hu et al., 2024). However, its application in pruning LLMs has not been well explored. In this paper, we derive the use of the representation mutual information metric to efficiently and rapidly determine the relative importance of each layer in LLMs during sparse fine-tuning. Through extensive experiments, we validate the effectiveness of the representation mutual information metric in pruning LLMs.

**Low-Rank Adaptation (LoRA).** LoRA (Hu et al., 2021) stands out as a highly effective parameter-efficient fine-tuning (PEFT) method (Houlsby et al., 2019; Pfeiffer et al., 2020; Lester et al., 2021; Liu et al., 2021), which incorporates trainable low-rank matrices that seamlessly reintegrate into the original model weights post-tuning, ensuring maintained efficiency without added latency or memory overhead. In LoRA fine-tuning, a crucial rank parameter dictates the tuning budget for each layer. AdaLoRA (Zhang et al., 2023b) underscores the importance of adaptive allocation, suggesting that this budget be tailored according to the significance score of each weight matrix. SoRA (Ding et al., 2023) is to dynamically adjust the rank of low-rank adaptation in the training process with a sparse gating unit trained by proximal gradient method. ALoRA (Liu et al., 2024) evaluates the importance of each rank, iteratively prunes low-contribution ranks, and reallocates resources to achieve dynamic adjustment of ranks. Similarly, we have identified the issue with the distribution of fine-tuning parameters during the fine-tuning of sparse LLMs. Uniformly setting the rank size like LoRA, does not effectively restore the accuracy of sparse LLMs. Therefore, this paper advocates for the dynamic allocation of the rank parameter budget, based on the sparse reconstruction errors across different layers, to optimize tuning efficacy. Although both LoSA and previous related works propose adjusting the rank in LoRA to achieve efficient parameter allocation, LoSA's dynamic rank adjustment strategy is specifically designed for sparse LLMs. Allocating fine-tuning parameters based on reconstruction error helps minimize the reconstruction error of sparse LLMs.

**Joint Sparsity and LoRA.** Combining network sparsity with LoRA has been shown to effectively enhance the accuracy of sparse LLMs (Li et al., 2024b;a; Zhao et al., 2024). For instance, LLM-Pruner (Ma et al., 2023) executes a one-shot structured pruning of LLMs, followed by fine-tuning using LoRA. LoRAPrune (Zhang et al., 2023a) implements iterative structured pruning, where weight importance is determined by replacing gradients on full weights with those calculated via LoRA. LoSparse (Li et al., 2023a) performs structured pruning on LLMs, using a combination of low-rank and sparse matrices to approximate the original weight matrix. LoRAShear (Chen et al., 2023) utilizes LoRA in conjunction with dynamic fine-tuning strategies to reinstate knowledge in structural pruning LLMs. All these studies apply LoRA to fine-tune structural pruning LLMs. Adjusting the input/output dimensions of the two low-rank adaptations in LoRA and integrating them into the structural pruning weights is straightforward (Zhao et al., 2024; Guo et al., 2023). However, this approach is not viable for unstructured pruning (network sparsity). Unstructured pruning removes individual weights, resulting in sparse LLMs. In contrast, low-rank adaptations remain dense even after dimensional adjustments, making it impossible to merge them into sparse LLMs. Consequently, this paper aims to explore effective techniques for integrating low-rank adaptations into

Table 10: Experimental results of comparison between cubic and linear sparsity schedule.

| Method | Perplexity | Accuracy |
|--------|-----------|----------|
| Linear | 8.04 | 57.79 |
| **Cubic** | **7.88** | **58.06** |

the sparse weights of LLM. The goal is to ensure that sparse LLMs and low-rank adaptations share the same sparse mask, thereby the model's sparsity is preserved and inference latency remains unaffected.

## C  MORE ABLATION STUDIES

### C.1  CUBIC VS. LINEAR SPARSITY SCHEDULE

In Section 2.4, we gradually increase the sparsity rate using cubic sparsity schedule, where we compare this with the setting of linearly increasing the sparsity rate. Linear sparsity schedule can be be expressed as:

$$\Theta^t = \Theta^f - \Theta^f \left( 1 - \frac{t}{T} \right), \quad t = 1, 2, ..., T \tag{10}$$

where $\Theta^f$ is final sparsity rate and $\Theta^t$ denotes the average sparsity rate of the $n$ layers at step $t$.

We present the impact of using a cubic sparsity schedule versus a linear sparsity schedule on final accuracy in Table 10. Specifically, we provide results of LoSA fine-tuning LLaMA-2-7B at a 60% sparsity rate, as obtained using Wanda method. The cubic sparsity schedule consistently outperforms the linear sparsity schedule in terms of accuracy. Compared to the linear sparsity schedule, which removes redundant connections uniformly, the cubic sparsity schedule prunes the network more aggressively in the initial phase when redundant connections are abundant, and then gradually reduces the number of weights pruned each time as fewer weights remain in the network. Our experiments showed that the cubic sparsity schedule performed better than the linear sparsity schedule, which is why we adopted it.

### C.2  PARTIAL OR ALL LINEAR LAYERS?

In our experiments, we applied low-rank adaptation to all linear layers in both the attention and MLP modules of the LLM. We present the experimental results of applying low-rank adaptation to only a subset of the linear layers in Table 11. Specifically, we experimented with applying low-rank adaptation solely to the linear layers within the attention of 60% sparse LLaMA-2-7B, as obtained using Wanda. The results indicated that partial application of low-rank adaptation yielded worse performance compared to applying it to all linear layers. We believe that since all the linear layers are sparse, low-rank adaptation should be added to all of them to maximize the recovery of accuracy.

Table 11: Experimental results of adding low-rank adaptation to partial linear layers.

| Linear Layer | Perplexity | Accuracy |
|---|---|---|
| Q,V | 8.22 | 56.59 |
| Q,K,V,O | 8.13 | 56.95 |
| **All** | **7.88** | **58.06** |

### C.3  COMPARISON WITH STRUCTURED PRUNING.

We compare the performance of structured pruning and unstructured pruning in Table 12. Specifically, for structured pruning, we use the LLM-Pruner (Ma et al., 2023) method to obtain the pruned LLM and then perform LoRA fine-tuning on the pruned LLM, with the fine-tuning settings referenced in Section 3.1. The acceleration data for structured pruning is obtained from LLM-Pruner paper. From the experimental data in the table, we can see that unstructured pruning can achieve basically the same acceleration effect on the GPU as structured pruning, while maintaining significantly better accuracy than structured pruning.

### C.4  LoSA VS. "FINE-TUNE FIRST, THEN SPARSIFY"

A key issue addressed by LoSA is that LoRA cannot be merged into sparse LLMs. However, there is a simple solution to this problem: first fine-tune the LLM using LoRA, and then apply sparsification methods such as SparseGPT or Wanda. We call the above method as "Fine-tune first, then sparsify".

Table 12: Comparison results of unstructured pruning and structured pruning.

| Method | Sparsity | Perplexity | Accuracy | Speedup |
|---|---|---|---|---|
| LLaMA-2-7B | 0% | 5.12 | 61.88 | 1.00× |
| LLM-Pruner w. LoRA | Structured 50% | 23.25 | 51.02 | **1.85** × |
| **SparseGPT w. LoSA** | **Unstructured 50%** | **6.25** | **61.32** | 1.71 × |

We compare LoSA with this "Fine-tune first, then sparsify" approach in Table 13. We use the Wanda method to obtain sparse LLaMA-2-7B model and use LoSA or the "Fine-tune first, then sparsify" method to fine-tune. As shown in the experimental data from Table 13, the accuracy of the "Fine-tune first, then sparsify" method is lower than that of LoSA, especially in high sparsity settings, where it leads to severe accuracy degradation. In contrast, LoSA adopts an iterative sparse fine-tuning approach that maintains good accuracy and can be merged into sparse LLMs, further demonstrating the superiority of the LoSA method.

Table 13: LoSA vs. "Fine-tune first, then sparsify".

| Method | Sparsity | Perplexity | Accuracy |
|---|---|---|---|
| LLaMA-2-7B | 0 % | 5.12 | 61.88 |
| "Fine-tune first, then sparsify" | 50 % | 7.00 | 60.35 |
| **LoSA** | **50** % | **6.21** | **60.85** |
| "Fine-tune first, then sparsify" | 60 % | 11.00 | 53.44 |
| **LoSA** | **60** % | **7.88** | **58.06** |
| "Fine-tune first, then sparsify" | 70 % | 77.57 | 34.99 |
| **LoSA** | **70** % | **10.94** | **51.65** |

## C.5 LoSA vs. Sparse LoRA

We show the effect of fine-tuning the sparse LLMs using Sparse LoRA, where Sparse LoRA is an improved version of LoRA that has the same mask as sparse LLMs and can be merged into sparse LLMs. Specifically, we show the results for the 70% sparse LLaMA-2-7B (Touvron et al., 2023b) model in Table 14.

Table 14: LoSA vs. Sparse LoRA.

| Method | Perplexity | HellaSwag | Winogrande | BoolQ | OBQA | PIQA | ARC-e | ARC-c | Mean |
|---|---|---|---|---|---|---|---|---|---|
| LLaMA-2-7B | 5.12 | 57.17 | 68.90 | 77.74 | 31.40 | 78.07 | 76.39 | 43.52 | 61.88 |
| SparseGPT | 27.42 | 33.08 | 58.41 | 64.89 | 17.40 | 62.46 | 43.22 | 22.01 | 43.07 |
| w. Sparse LoRA | 11.26 | 43.63 | 62.06 | 63.46 | 22.80 | 70.84 | 57.22 | 29.01 | 49.86 |
| w. LoRA | 11.06 | 44.80 | 62.90 | 63.36 | 24.20 | 71.22 | 58.71 | 30.12 | 50.76 |
| **w.LoSA** | **10.82** | **46.06** | **63.85** | **70.15** | **24.80** | **71.93** | **60.44** | **30.35** | **52.51** |
| Wanda | 79.67 | 27.92 | 49.33 | 52.87 | 12.60 | 55.33 | 30.60 | 18.69 | 35.33 |
| w. Sparse LoRA | 12.74 | 40.53 | 56.84 | 64.08 | 22.20 | 68.53 | 55.77 | 26.37 | 47.76 |
| w. LoRA | 12.57 | 40.77 | 57.22 | 64.19 | 22.40 | 68.55 | 57.32 | 26.79 | 48.18 |
| **w. LoSA** | **10.94** | **45.10** | **60.93** | **67.65** | **25.20** | **71.06** | **62.50** | **29.10** | **51.65** |

Since Sparse LoRA's Low-rank adaptation is also sparse, its accuracy is worse than LoRA, and it is also much worse than LoSA. Additionally, since both Sparse LoRA and LoSA can be merged into sparse LLMs, their inference acceleration effects are basically the same, and both outperform LoRA. Overall, our proposed LoSA method outperforms both Sparse LoRA and LoRA in terms of accuracy and inference acceleration.

## C.6 ABLATION EXPERIMENT ON OPT MODEL.

We present ablation experiments of our proposed strategies on the LLaMA-2-7B model in Section 3.5. In this section, we further demonstrate the effectiveness of our two proposed strategies, including Layer-wise Sparsity Rate (**LSR** and Section 2.2) and Sparsity-Aware Rank Allocation (**SRA, Section 2.3**), on the OPT model which is non-LLaMA architecture. Specifically, we used the Wanda method to obtain a 70% sparse OPT-13B (Zhang et al., 2022) model. The experimental results are in Table 15.

Table 15: Ablation experiment results on the OPT model.

| Method | Perplexity | HellaSwag | Winogrande | BoolQ | OBQA | PIQA | ARC-e | ARC-c | Mean |
|---|---|---|---|---|---|---|---|---|---|
| **LoSA** | **19.75** | **45.20** | **59.91** | **60.96** | **24.80** | **73.39** | **57.65** | **29.01** | **50.13** |
| w/o LSR | 20.72 | 44.21 | 59.32 | 59.34 | 24.20 | 72.45 | 57.10 | 28.15 | 49.24 |
| w/o SRA | 20.55 | 44.84 | 59.66 | 59.24 | 24.40 | 72.69 | 57.07 | 28.41 | 49.47 |
| w/o LSR & SRA | 21.48 | 43.35 | 58.78 | 58.65 | 23.90 | 72.09 | 56.45 | 27.56 | 48.68 |

We can see that removing either the LSR or SRA leads to a decrease in LoSA accuracy. The results demonstrate the soundness and effectiveness of LSR and SRA across different architectures.

## D EXTENDING LOSA TO STRUCTURED PRUNING

Although LoSA focuses on fine-tuning unstructured pruned LLMs, we extended the LoSA method to fine-tune structured pruned LLMs in this section. We use the Wanda-sp (An et al., 2024) method to determine the mask of structured pruned LLMs. We compared LoSA with SliceGPT (Ashkboos et al., 2024), LLM-Pruner (Ma et al., 2023), LoRAPrune (Zhang et al., 2023a) and LoRAShear (Chen et al., 2023) on the LLaMA-1-7B model (Touvron et al., 2023a) with 20% pruning rate. The experimental results are reported in Table 16.

Table 16: Experimental results of LoSA fine-tuning structured pruned LLMs.

| Method | Perplexity | HellaSwag | Winogrande | BoolQ | OBQA | PIQA | ARC-e | ARC-c | Mean |
|---|---|---|---|---|---|---|---|---|---|
| LLaMA-1-7B | 5.69 | 73.18 | 78.35 | 72.99 | 67.01 | 67.45 | 41.38 | 42.40 | 63.25 |
| SliceGPT | 8.71 | 37.89 | 64.09 | 45.67 | 62.75 | 53.62 | 31.74 | 33.20 | 46.99 |
| LLM-pruner | 8.14 | 69.54 | 76.44 | 68.11 | 65.11 | 63.43 | 37.88 | 40.00 | 60.07 |
| LoRAPrune | 7.63 | 65.82 | 79.31 | 70.00 | 62.76 | 65.87 | 37.69 | 39.14 | 60.05 |
| LoRAShear | / | 70.17 | 76.89 | 68.69 | 65.83 | 64.11 | 38.77 | 39.97 | 60.63 |
| **LoSA** | **7.07** | **71.67** | **78.17** | **71.56** | **65.86** | **66.93** | **40.66** | **40.50** | **62.19** |

Our LoSA method outperforms SliceGPT, LLM-Pruner, LoRAPrune and LoRAShear as shown above, demonstrating the superior performance of LoSA.

## E EXTENDING LOSA TO FINE-TUNE OTHER TRAINING-FREE SPARSITY METHODS

Our LoSA method is designed for fine-tuning sparse LLMs, which means that LoSA can be combined with any training-free sparsity methods to enhance their accuracy. We have demonstrated the improvement that LoSA brings to SparseGPT (Frantar & Alistarh, 2023) and Wanda (Sun et al., 2023). We further show the performance improvements of LoSA on other training-free sparsity methods, including Pruner-Zero (Dong et al., 2024) and ALPS (Meng et al., 2024). All experimental data are based on a 70% sparse LLaMA-2-7B (Touvron et al., 2023b) model. The experimental results are reported in Table 17.

From the data above, we can observe that the accuracy of training-free sparse LLMs has significantly decreased compared to the dense model. LoSA effectively improves the accuracy of sparse LLMs, outperforming LoRA.

Table 17: Experimental results of LoSA fine-tuning other training-free sparsity methods.

| Method | Perplexity | HellaSwag | Winogrande | BoolQ | OBQA | PIQA | ARC-e | ARC-c | Mean |
|---|---|---|---|---|---|---|---|---|---|
| LLaMA-2-7B | 5.12 | 57.17 | 68.90 | 77.74 | 31.40 | 78.07 | 76.39 | 43.52 | 61.88 |
| Pruner-Zero | 103.15 | 27.56 | 50.99 | 41.93 | 13.00 | 56.90 | 34.47 | 18.60 | 34.78 |
| w. LoRA | 11.56 | 43.43 | 60.46 | 67.19 | 21.00 | 70.40 | 59.60 | 27.47 | 49.94 |
| **w. LoSA** | **10.78** | **45.56** | **62.10** | **69.15** | **25.00** | **71.73** | **61.08** | **29.45** | **52.01** |
| ALPS | 19.31 | 38.35 | 61.96 | 64.59 | 22.20 | 66.82 | 48.37 | 24.95 | 46.75 |
| w. LoRA | 10.83 | 47.54 | 62.88 | 69.11 | 27.00 | 73.23 | 61.70 | 29.78 | 53.03 |
| **w. LoSA** | **10.28** | **49.90** | **64.34** | **71.38** | **28.10** | **75.24** | **63.78** | **31.27** | **54.86** |

## F  COMPARISON OF LoSA AND MORE BASELINES

We present a comparison of LoSA with AdaLoRA (Zhang et al., 2023b) and SoRA (Ding et al., 2023) which dynamically adjust the rank of LoRA like LoSA. Since AdaLoRA and SoRA have only been experimented on smaller models and there is no experimental data for LLMs such as LLaMA, we use the open source codes of AdaLoRA and SoRA to fine-tune a 70% sparse LLaMA-2-7B (Touvron et al., 2023b) model obtained by the Wanda (Sun et al., 2023) method. Since the rank of LoRA is 8, according to the original paper, the initial rank for each incremental matrix in AdaLoRA is 12. The rank of the SoRA method is set to 8, and other hyperparameters are set according to the original paper. Other experimental settings follow those in Section 3.1 and are aligned with the settings of LoRA and LoSA. The experimental results are reported in Table 18.

Table 18: Comparison of AdaLoRA (Zhang et al., 2023b) SoRA (Ding et al., 2023) and LoSA.

| Method | Perplexity | HellaSwag | Winogrande | BoolQ | OBQA | PIQA | ARC-e | ARC-c | Mean |
|---|---|---|---|---|---|---|---|---|---|
| LLaMA-2-7B | 5.12 | 57.17 | 68.90 | 77.74 | 31.40 | 78.07 | 76.39 | 43.52 | 61.88 |
| Wanda | 79.67 | 27.92 | 49.33 | 52.87 | 12.60 | 55.33 | 30.60 | 18.69 | 35.33 |
| w. LoRA | 12.57 | 40.77 | 57.22 | 64.19 | 22.40 | 68.55 | 57.32 | 26.79 | 48.18 |
| w. AdaLoRA | 12.08 | 41.01 | 57.78 | 64.73 | 23.00 | 69.09 | 57.77 | 26.90 | 48.61 |
| w. SoRA | 11.89 | 41.37 | 57.87 | 64.95 | 23.40 | 68.78 | 58.25 | 27.17 | 48.83 |
| **w. LoSA** | **10.94** | **45.10** | **60.93** | **67.65** | **25.20** | **71.06** | **62.50** | **29.10** | **51.65** |

The results clearly show that our LoSA method outperforms AdaLoRA and SoRA, demonstrating the effectiveness of LoSA. This is evident because AdaLoRA and SoRA only dynamically adjust the rank, and the weights of AdaLoRA and SoRA cannot be merged into sparse LLMs. In contrast, LoSA dynamically adjusts the rank based on reconstruction error, determines layer-wise sparsity rates for sparse LLMs, and adopts dynamic sparse fine-tuning. Additionally, LoSA weights can be merged into sparse LLMs. These strategies ensure that LoSA achieves better accuracy than AdaLoRA and SoRA.

## G  MORE EXPERIMENTAL RESULTS OF N:M SPARSITY

We demonstrated the accuracy of LoSA fine-tuning sparse LLMs with mixed 2:8 sparsity in Section 3.4. In this section, we further demonstrate the accuracy of LoSA fine-tuning sparse LLMs with mixed 2:4 sparsity and show the acceleration effect of the sparse LLMs with mixed 2:4 and mixed 2:8 sparsity on GPU. The results of LoSA fine-tuning the mixed 2:4 sparse LLaMA-2-7B (Touvron et al., 2023b) obtained by Wanda (Sun et al., 2023) method are shown in Table 19.

LoSA improves accuracy for mixed 2:4 sparsity and outperforms LoRA. Since mixed 2:4 and mixed 2:8 sparsity is a specific type of sparsity pattern, it can also leverage the nm-vllm (NeuralMagic, 2024) inference engine to achieve accelerated inference on GPUs. We also measured the inference acceleration effect of sparse LLaMA-2-7B with mixed 2:4 and mixed 2:8 sparsity on NVIDIA RTX 4090 24GB GPU. The results are shown in Table 20.

Table 19: Experimental results of LoSA fine-tuning the mixed 2:4 sparse LLaMA-2-7B obtained by Wanda method.

| Method | Sparsity | Perplexity | HellaSwag | Winogrande | BoolQ | OBQA | PIQA | ARC-e | ARC-c | Mean |
|---|---|---|---|---|---|---|---|---|---|---|
| LLaMA-2-7B | 0% | 5.12 | 57.17 | 68.90 | 77.74 | 31.40 | 78.07 | 76.39 | 43.52 | 61.88 |
| Wanda | 2:4 | 11.02 | 40.92 | 62.43 | 67.65 | 24.20 | 70.84 | 61.78 | 31.20 | 51.29 |
| w. LoRA | 2:4 | 8.27 | 50.37 | 64.80 | 72.81 | 27.60 | 75.19 | 69.40 | 35.58 | 56.54 |
| **w. LoSA** | **Mixed 2:4** | **7.72** | **51.85** | **66.01** | **74.51** | **29.70** | **76.54** | **71.08** | **37.26** | **58.14** |

Table 20: Speedup of LLaMA-2-7B with mixed N:M sparsity on GPU.

| Speed | Dense | Mixed 2:4 | Mixed 2:8 |
|---|---|---|---|
| Throughput (tokens/s) | 57.35 | 98.35 | 133.40 |
| Speedup | 1.00× | 1.71× | 2.33× |

## H  EFFICIENCY ANALYSIS AS THE MODEL SIZE INCREASES

We measured the time consumption of our proposed methods, Layer-wise Sparsity Rate (**LSR, Section 2.2**) and Sparsity-Aware Rank Allocation (**SRA, Section 2.3**), on LLMs of different parameter sizes using a single NVIDIA A100 80GB GPU. The results are reported in Table 21.

Table 21: Time consumption of our proposed methods on models with different parameter sizes.

| Time (seconds) | 7B | 8B | 13B | 30B | 70B |
|---|---|---|---|---|---|
| LSR | 48 | 34 | 74 | 140 | 332 |
| SRA | 46 | 32 | 71 | 132 | 321 |

From the table, we can observe that as the model size increases, the required computation time also increases. However, for the largest 70B model, the computation time for LSR and SRA are only 332 seconds and 321 seconds, respectively, which are very fast and have minimal computational overhead.

## I  ANALYSIS OF SPARSITY RATE

We plot the layer-wise sparsity rate of the sparse LLMs obtained by our LoSA method in Figure 3.

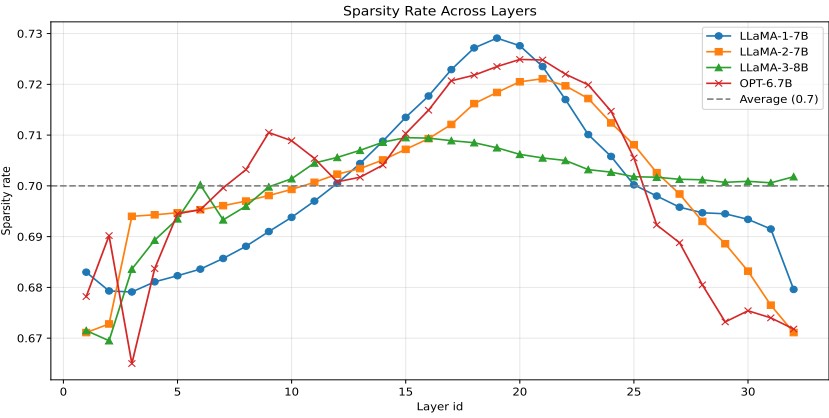

Figure 3: Layer-wise sparsity rates of different LLMs.

We observed that the RMI metric tends to assign lower sparsity rates to the initial and final layers of LLMs while allocating higher sparsity rates to the middle layers. From the results, we can see that there is a lot of redundancy in the middle layer of LLMs.

## J  DETAILED ZERO-SHOT TASK RESULTS

We evaluated a series of zero-shot learning tasks, as shown in Tables 2 and 3. These tasks include HellaSwag (Zellers et al., 2019), Winogrande (Sakaguchi et al., 2021), BoolQ (Clark et al., 2019), OpenBookQA (Mihaylov et al., 2018), PIQA (Bisk et al., 2020), ARC-Easy, and ARC-Challenge (Clark et al., 2018). We present detailed task performance metrics in Tables 22, 23, 24, 25 and 26, providing a comprehensive understanding of the zero-shot capabilities of the related models.

Table 22: Zero-shot accuracy results of `LoSA` for sparse LLaMAs at 50% sparsity.

| Model | Method | HellaSwag | Winogrande | BoolQ | OBQA | PIQA | ARC-e | ARC-c | Mean |
|---|---|---|---|---|---|---|---|---|---|
| LLaMA-1-7B | Dense | 56.92 | 69.93 | 75.05 | 34.40 | 78.67 | 75.34 | 41.89 | 61.74 |
| | SparseGPT | 51.43 | 67.88 | 72.05 | 30.00 | 75.30 | 71.38 | 37.71 | 57.96 |
| | w. LoRA | 55.72 | 68.59 | 74.16 | 31.00 | 76.39 | 71.46 | **41.47** | 59.83 |
| | **w. LoSA** | **55.72** | **69.22** | **76.06** | **32.00** | **77.04** | **72.90** | 40.87 | **60.54** |
| | Wanda | 51.85 | 66.06 | 71.22 | 28.80 | 75.63 | 69.11 | 36.86 | 57.08 |
| | w. LoRA | 55.39 | 67.01 | 72.72 | 32.00 | 77.11 | 72.35 | 39.68 | 59.46 |
| | **w. LoSA** | **55.72** | **68.03** | **73.36** | 31.40 | **77.15** | **73.48** | **41.21** | **60.09** |
| LLaMA-1-13B | Dense | 59.94 | 72.77 | 77.89 | 33.20 | 79.16 | 77.40 | 46.50 | 63.84 |
| | SparseGPT | 54.95 | 71.67 | 76.97 | 31.20 | 77.26 | 72.47 | 41.98 | 60.93 |
| | w. LoRA | 58.90 | 69.93 | 80.00 | 33.40 | 78.67 | 73.93 | 43.88 | 62.68 |
| | **w. LoSA** | **59.29** | **71.19** | **80.03** | **33.60** | **79.22** | **73.95** | **44.37** | **63.09** |
| | Wanda | 55.71 | 71.98 | 75.90 | 32.20 | 77.26 | 73.19 | 43.52 | 61.39 |
| | w. LoRA | 58.64 | 71.03 | 79.69 | 34.50 | 78.40 | 74.41 | 44.79 | 63.07 |
| | **w. LoSA** | **58.85** | **71.82** | **79.77** | **34.60** | **78.78** | **74.49** | **46.08** | **63.49** |
| LLaMA-1-30B | Dense | 63.35 | 75.69 | 82.69 | 36.00 | 81.01 | 80.30 | 52.82 | 67.41 |
| | SparseGPT | 59.15 | 75.22 | 82.32 | 35.00 | 78.20 | 78.96 | 48.56 | 65.34 |
| | w. LoRA | 60.50 | 75.34 | 83.46 | 35.20 | 79.22 | 80.05 | 50.60 | 66.31 |
| | **w. LoSA** | **61.16** | **75.43** | **83.55** | **35.30** | **79.81** | **80.32** | **52.45** | **66.86** |
| | Wanda | 60.93 | 73.48 | 81.90 | 34.60 | 79.27 | 79.29 | 49.66 | 65.59 |
| | w. LoRA | 62.37 | 73.72 | 82.72 | **36.60** | 79.54 | 79.80 | 49.83 | 66.37 |
| | **w. LoSA** | **63.02** | **74.35** | **85.20** | 35.20 | **79.71** | **80.22** | **52.65** | **67.19** |
| LLaMA-2-7B | Dense | 57.17 | 68.90 | 77.74 | 31.40 | 78.07 | 76.39 | 43.52 | 61.88 |
| | SparseGPT | 52.37 | 69.85 | 75.02 | 29.20 | 75.46 | 73.27 | 39.85 | 59.29 |
| | w. LoRA | 55.63 | 68.35 | 76.94 | 31.90 | **76.09** | 73.32 | 41.04 | 60.47 |
| | **w. LoSA** | **55.73** | **68.67** | **77.19** | **32.00** | 75.71 | **77.42** | **42.49** | **61.32** |
| | Wanda | 52.49 | 68.19 | 75.99 | 31.20 | 76.00 | 72.77 | 39.59 | 59.46 |
| | w. LoRA | 55.31 | 68.11 | 77.06 | 31.80 | 77.58 | 72.73 | 40.96 | 60.51 |
| | **w. LoSA** | **55.75** | **68.13** | **77.13** | **33.00** | **77.59** | **73.06** | **41.21** | **60.85** |
| LLaMA-2-13B | Dense | 60.06 | 72.22 | 80.52 | 35.20 | 79.11 | 79.42 | 48.46 | 65.00 |
| | SparseGPT | 55.83 | 72.77 | 81.44 | 32.60 | 78.02 | 74.83 | 42.24 | 62.53 |
| | w. LoRA | 58.68 | 72.06 | 80.58 | 34.40 | 78.78 | 77.48 | 45.33 | 63.90 |
| | **w. LoSA** | **58.92** | **72.30** | **81.44** | **35.00** | **79.38** | **77.68** | **45.88** | **64.23** |
| | Wanda | 56.90 | 71.35 | 81.84 | 32.00 | 78.40 | 76.18 | 43.52 | 62.88 |
| | w. LoRA | 58.77 | 71.51 | 81.77 | 33.20 | 79.43 | 76.89 | 45.31 | 63.84 |
| | **w. LoSA** | **59.00** | **71.59** | **81.83** | **33.80** | **79.47** | **77.57** | **47.01** | **64.32** |
| LLaMA-2-70B | Dense | 66.10 | 78.06 | 83.40 | 37.20 | 82.21 | 82.55 | 54.44 | 69.14 |
| | SparseGPT | 63.80 | **78.85** | 83.55 | 38.20 | 81.94 | 82.40 | 53.75 | 68.93 |
| | w. LoRA | 63.45 | 78.37 | 84.74 | 39.20 | **82.97** | 83.08 | 55.12 | 69.56 |
| | **w. LoSA** | **64.06** | 78.57 | **85.34** | 39.20 | 82.54 | **83.31** | **55.20** | **69.82** |
| | Wanda | 64.10 | 78.14 | 82.50 | 37.40 | 81.88 | 80.80 | 52.65 | 68.21 |
| | w. LoRA | 64.06 | 76.64 | 83.30 | 38.10 | 82.54 | **83.38** | 55.29 | 69.04 |
| | **w. LoSA** | **64.16** | 77.74 | **85.57** | **38.20** | **82.59** | 83.16 | **56.14** | **69.65** |
| LLaMA-3-8B | Dense | 60.19 | 72.77 | 81.35 | 34.80 | 79.71 | 80.09 | 50.43 | 65.62 |
| | SparseGPT | 53.39 | 72.38 | 79.27 | 30.80 | 76.06 | 73.02 | 41.98 | 60.99 |
| | w. LoRA | 57.40 | 71.82 | 81.91 | 31.40 | 78.51 | 76.22 | 45.14 | 63.20 |
| | **w. LoSA** | **57.85** | **72.77** | **81.99** | **32.80** | **78.62** | **76.98** | **48.38** | **64.20** |
| | Wanda | 51.23 | 70.24 | 78.69 | 30.20 | 75.68 | 71.04 | 40.44 | 59.65 |
| | w. LoRA | 56.95 | 72.22 | 78.18 | 31.20 | 78.18 | 76.01 | 45.82 | 62.65 |
| | **w. LoSA** | **57.29** | **72.38** | **78.75** | **32.00** | **78.45** | **76.52** | **48.21** | **63.20** |
| LLaMA-3.1-8B | Dense | 59.98 | 73.32 | 82.05 | 33.20 | 79.98 | 81.57 | 51.45 | 65.93 |
| | SparseGPT | 53.62 | 72.14 | 81.19 | 29.20 | 76.17 | 74.49 | 41.72 | 61.22 |
| | w. LoRA | 57.46 | 71.82 | 82.37 | 32.00 | 79.33 | 78.54 | 48.29 | 64.25 |
| | **w. LoSA** | **57.65** | **72.30** | **82.54** | **32.40** | **79.76** | **78.75** | **48.72** | **64.59** |
| | Wanda | 51.19 | 70.72 | 78.62 | 26.80 | 75.08 | 73.11 | 41.55 | 59.58 |
| | w. LoRA | 56.67 | 71.19 | 79.02 | 30.60 | 78.51 | 77.48 | 47.89 | 63.05 |
| | **w. LoSA** | **57.17** | **71.35** | **80.55** | **31.60** | **78.67** | **78.03** | **47.97** | **63.63** |

Table 23: Zero-shot accuracy results of `LoSA` for sparse LLaMAs at 60% sparsity.

| Model | Method | HellaSwag | Winogrande | BoolQ | OBQA | PIQA | ARC-e | ARC-c | Mean |
|---|---|---|---|---|---|---|---|---|---|
| LLaMA-1-7B | Dense | 56.92 | 69.93 | 75.05 | 34.40 | 78.67 | 75.34 | 41.89 | 61.74 |
| | SparseGPT | 44.86 | 63.61 | 70.24 | 24.40 | 73.10 | 62.62 | 30.20 | 52.72 |
| | w. LoRA | 52.27 | 65.82 | 64.92 | 29.20 | 75.84 | 67.30 | 36.12 | 55.92 |
| | **w. LoSA** | **52.30** | **65.85** | **74.43** | **29.20** | **76.06** | **67.55** | **36.26** | **57.38** |
| | Wanda | 43.63 | 62.04 | 67.19 | 25.00 | 73.02 | 62.61 | 30.34 | 51.98 |
| | w. LoRA | 52.00 | 63.85 | 66.67 | 29.40 | 75.48 | 66.25 | 34.13 | 55.39 |
| | **w. LoSA** | **52.16** | **64.17** | **69.66** | **29.70** | **75.76** | **66.41** | **35.58** | **56.21** |
| LLaMA-1-13B | Dense | 59.94 | 72.77 | 77.89 | 33.20 | 79.16 | 77.40 | 46.50 | 63.84 |
| | SparseGPT | 49.06 | 68.75 | 70.37 | 27.60 | 75.63 | 68.40 | 36.20 | 56.57 |
| | w. LoRA | 55.80 | 67.96 | 77.34 | 30.60 | 77.31 | 70.75 | 40.27 | 60.00 |
| | **w. LoSA** | **56.22** | **69.77** | **78.93** | **32.00** | **77.64** | **71.63** | **41.21** | **61.06** |
| | Wanda | 48.92 | 68.19 | 69.82 | 27.64 | 74.91 | 68.92 | 34.93 | 56.19 |
| | w. LoRA | 55.34 | 69.06 | 76.27 | 30.20 | 76.88 | 70.20 | 39.76 | 59.67 |
| | **w. LoSA** | **55.81** | **69.85** | **76.29** | **31.00** | **77.97** | **72.60** | **42.66** | **60.88** |
| LLaMA-1-30B | Dense | 63.35 | 75.69 | 82.69 | 36.00 | 81.01 | 80.30 | 52.82 | 67.41 |
| | SparseGPT | 55.03 | 72.80 | 76.50 | 32.20 | 76.83 | 74.71 | 43.32 | 61.63 |
| | w. LoRA | 61.01 | 72.69 | 81.25 | 35.00 | 78.51 | 78.07 | 47.78 | 64.90 |
| | **w. LoSA** | **62.03** | **72.91** | **81.67** | **36.30** | **79.32** | **79.25** | **50.32** | **65.97** |
| | Wanda | 56.71 | 72.30 | 76.24 | 31.60 | 77.67 | 76.19 | 46.52 | 62.46 |
| | w. LoRA | 60.31 | 72.65 | 79.05 | 34.60 | 78.89 | 77.82 | 48.21 | 64.50 |
| | **w. LoSA** | **61.02** | **72.75** | **81.56** | **36.40** | **79.22** | **79.12** | **50.94** | **65.86** |
| LLaMA-2-7B | Dense | 57.17 | 68.90 | 77.74 | 31.40 | 78.07 | 76.39 | 43.52 | 61.88 |
| | SparseGPT | 45.74 | 65.90 | 71.99 | 25.80 | 71.11 | 64.02 | 32.76 | 53.90 |
| | w. LoRA | 51.51 | 65.04 | 73.79 | 30.00 | 74.79 | 68.82 | 38.57 | 57.50 |
| | **w. LoSA** | **52.95** | **67.32** | **73.82** | **31.00** | **75.97** | **69.95** | **38.63** | **58.52** |
| | Wanda | 44.22 | 64.88 | 65.84 | 25.20 | 72.09 | 64.56 | 30.80 | 52.51 |
| | w. LoRA | 51.28 | 65.82 | 70.43 | 30.20 | 74.97 | 69.36 | 35.75 | 56.83 |
| | **w. LoSA** | **51.62** | **66.93** | **74.04** | **31.40** | **74.98** | **70.37** | **37.12** | **58.06** |
| LLaMA-2-13B | Dense | 60.06 | 72.22 | 80.52 | 35.20 | 79.11 | 79.42 | 48.46 | 65.00 |
| | SparseGPT | 49.89 | 70.88 | 77.28 | 28.80 | 75.41 | 68.98 | 36.18 | 58.20 |
| | w. LoRA | 55.85 | 69.01 | 78.13 | 32.40 | 77.42 | 74.03 | 40.70 | 61.08 |
| | **w. LoSA** | **56.13** | **69.69** | **79.05** | **32.60** | **77.99** | **74.74** | **41.47** | **61.67** |
| | Wanda | 48.82 | 68.75 | 77.28 | 29.00 | 75.84 | 69.87 | 37.80 | 58.19 |
| | w. LoRA | 55.61 | 68.98 | 78.29 | 31.20 | 77.42 | 72.35 | 42.49 | 60.91 |
| | **w. LoSA** | **55.90** | **69.93** | **78.38** | **32.40** | **77.91** | **74.54** | **43.69** | **61.82** |
| LLaMA-2-70B | Dense | 66.10 | 78.06 | 83.40 | 37.20 | 82.21 | 82.55 | 54.44 | 69.14 |
| | SparseGPT | 59.41 | 76.64 | 83.85 | 35.60 | 80.35 | 80.26 | 49.49 | 66.52 |
| | w. LoRA | 61.86 | **77.74** | 84.83 | 38.00 | 81.66 | 81.61 | 52.56 | 68.32 |
| | **w. LoSA** | **62.23** | 77.43 | **84.90** | **38.40** | **82.69** | **82.90** | **54.70** | **69.04** |
| | Wanda | 59.43 | 76.16 | 84.10 | 36.20 | 79.92 | 80.09 | 47.95 | 66.27 |
| | w. LoRA | 62.43 | 76.01 | 84.25 | 38.50 | 81.01 | 82.20 | 53.24 | 68.24 |
| | **w. LoSA** | **62.54** | **76.72** | **86.48** | **38.60** | **81.07** | **83.04** | **55.12** | **69.08** |
| LLaMA-3-8B | Dense | 60.19 | 72.77 | 81.35 | 34.80 | 79.71 | 80.09 | 50.43 | 65.62 |
| | SparseGPT | 45.84 | 68.51 | 77.77 | 22.80 | 70.57 | 62.16 | 30.72 | 54.05 |
| | w. LoRA | 52.91 | 68.67 | 76.91 | 26.20 | 75.17 | 72.22 | 40.78 | 58.98 |
| | **w. LoSA** | **53.48** | **70.24** | **80.37** | **28.20** | **75.19** | **72.64** | **41.98** | **60.30** |
| | Wanda | 38.02 | 60.14 | 68.56 | 20.00 | 67.95 | 59.93 | 27.73 | 48.90 |
| | w. LoRA | 50.32 | 65.43 | 73.76 | 26.80 | 74.37 | 69.44 | 39.51 | 57.09 |
| | **w. LoSA** | **51.54** | **68.51** | **75.57** | **27.00** | **75.57** | **71.42** | **40.61** | **58.60** |
| LLaMA-3.1-8B | Dense | 59.98 | 73.32 | 82.05 | 33.20 | 79.98 | 81.57 | 51.45 | 65.93 |
| | SparseGPT | 45.53 | 68.75 | 77.65 | 24.20 | 71.38 | 68.10 | 34.98 | 55.80 |
| | w. LoRA | 53.21 | 68.35 | 78.75 | 28.00 | 75.79 | 73.91 | 43.34 | 60.19 |
| | **w. LoSA** | **53.62** | **69.22** | **79.97** | **28.10** | **76.17** | **74.66** | **43.47** | **60.74** |
| | Wanda | 38.75 | 60.85 | 70.67 | 21.40 | 69.59 | 60.19 | 27.05 | 49.78 |
| | w. LoRA | 51.47 | 65.90 | 73.85 | 27.80 | 75.24 | 70.29 | 38.74 | 57.61 |
| | **w. LoSA** | **51.85** | **67.40** | **74.83** | **28.80** | **75.34** | **72.01** | **40.27** | **58.64** |

Table 24: Zero-shot accuracy results of LoSA for sparse LLaMAs at 70% sparsity.

| Model | Method | HellaSwag | Winogrande | BoolQ | OBQA | PIQA | ARC-e | ARC-c | Mean |
|---|---|---|---|---|---|---|---|---|---|
| LLaMA-1-7B | Dense | 56.92 | 69.93 | 75.05 | 34.40 | 78.67 | 75.34 | 41.89 | 61.74 |
| | SparseGPT | 34.58 | 56.43 | 64.80 | 16.80 | 64.25 | 45.24 | 23.12 | 43.60 |
| | w. LoRA | 45.86 | 60.93 | 63.12 | 23.20 | 70.62 | 58.67 | 30.46 | 50.41 |
| | **w. LoSA** | **46.45** | **64.09** | **68.65** | **24.80** | **72.14** | **60.77** | **32.25** | **52.74** |
| | Wanda | 28.86 | 52.80 | 59.69 | 14.20 | 57.56 | 31.27 | 17.75 | 37.45 |
| | w. LoRA | 41.70 | 57.85 | 65.05 | 22.60 | 69.48 | 54.12 | 27.30 | 48.30 |
| | **w. LoSA** | **45.13** | **60.06** | **67.65** | **24.40** | **71.65** | **59.72** | **29.78** | **51.20** |
| LLaMA-1-13B | Dense | 59.94 | 72.77 | 77.89 | 33.20 | 79.16 | 77.40 | 46.50 | 63.84 |
| | SparseGPT | 37.51 | 63.30 | 68.78 | 20.80 | 67.63 | 52.78 | 25.17 | 48.00 |
| | w. LoRA | 50.10 | 63.61 | 72.29 | 26.20 | 74.43 | 64.56 | 33.79 | 55.00 |
| | **w. LoSA** | **51.28** | **64.88** | **74.59** | **29.20** | **75.35** | **64.62** | **35.76** | **56.53** |
| | Wanda | 31.06 | 54.38 | 61.59 | 16.20 | 62.68 | 42.05 | 17.58 | 40.79 |
| | w. LoRA | 47.56 | 61.01 | 66.02 | 23.00 | 72.74 | 61.57 | 30.89 | 51.83 |
| | **w. LoSA** | **50.21** | **64.95** | **71.16** | **25.60** | **74.10** | **66.16** | **34.98** | **55.31** |
| LLaMA-1-30B | Dense | 63.35 | 75.69 | 82.69 | 36.00 | 81.01 | 80.30 | 52.82 | 67.41 |
| | SparseGPT | 44.56 | 69.30 | 65.35 | 25.80 | 72.42 | 65.78 | 32.25 | 53.64 |
| | w. LoRA | 55.66 | 72.13 | 79.36 | 30.40 | 77.15 | 73.86 | 42.83 | 61.63 |
| | **w. LoSA** | **56.32** | **73.40** | **80.09** | **32.30** | **77.42** | **73.94** | **43.10** | **62.37** |
| | Wanda | 44.23 | 67.01 | 66.70 | 26.40 | 72.03 | 64.86 | 32.25 | 53.35 |
| | w. LoRA | 53.78 | 67.72 | 77.74 | 30.80 | 76.44 | 68.22 | 38.74 | 59.06 |
| | **w. LoSA** | **56.21** | **69.77** | **78.50** | **32.20** | **77.31** | **73.90** | **43.09** | **61.57** |
| LLaMA-2-7B | Dense | 57.17 | 68.90 | 77.74 | 31.40 | 78.07 | 76.39 | 43.52 | 61.88 |
| | SparseGPT | 33.08 | 58.41 | 64.89 | 17.40 | 62.46 | 43.22 | 22.01 | 43.07 |
| | w. LoRA | 44.80 | 62.90 | 63.36 | 24.20 | 71.22 | 58.71 | 30.12 | 50.76 |
| | **w. LoSA** | **46.06** | **63.85** | **70.15** | **24.80** | **71.93** | **60.44** | **30.35** | **52.51** |
| | Wanda | 27.92 | 49.33 | 52.87 | 12.60 | 55.33 | 30.60 | 18.69 | 35.33 |
| | w. LoRA | 40.77 | 57.22 | 64.19 | 22.40 | 68.55 | 57.32 | 26.79 | 48.18 |
| | **w. LoSA** | **45.10** | **60.93** | **67.65** | **25.20** | **71.06** | **62.50** | **29.10** | **51.65** |
| LLaMA-2-13B | Dense | 60.06 | 72.22 | 80.52 | 35.20 | 79.11 | 79.42 | 48.46 | 65.00 |
| | SparseGPT | 36.90 | 61.64 | 66.02 | 21.00 | 67.57 | 52.61 | 25.94 | 47.38 |
| | w. LoRA | 49.86 | 66.77 | 71.99 | 26.40 | 74.21 | 63.97 | 32.94 | 55.16 |
| | **w. LoSA** | **50.57** | **67.56** | **76.42** | **28.20** | **74.27** | **67.47** | **35.67** | **57.16** |
| | Wanda | 29.60 | 51.70 | 62.32 | 13.60 | 58.65 | 37.21 | 19.11 | 38.88 |
| | w. LoRA | 45.70 | 60.93 | 62.20 | 24.00 | 71.98 | 60.23 | 29.27 | 50.62 |
| | **w. LoSA** | **46.79** | **62.90** | **68.20** | **25.20** | **73.65** | **63.38** | **30.80** | **53.00** |
| LLaMA-2-70B | Dense | 66.10 | 78.06 | 83.40 | 37.20 | 82.21 | 82.55 | 54.44 | 69.14 |
| | SparseGPT | 50.98 | **75.45** | 80.06 | 30.00 | 75.24 | 73.57 | 40.61 | 60.84 |
| | w. LoRA | 59.50 | 74.98 | 84.04 | 34.00 | 79.71 | 78.41 | 49.40 | 65.72 |
| | **w. LoSA** | **60.21** | 75.09 | **84.72** | **35.00** | **79.80** | **79.43** | **50.60** | **66.41** |
| | Wanda | 48.16 | 73.88 | 74.46 | 27.00 | 74.86 | 72.69 | 38.31 | 58.48 |
| | w. LoRA | 59.06 | **75.45** | 82.48 | 34.00 | 79.05 | 78.41 | 48.46 | 65.27 |
| | **w. LoSA** | **60.10** | 74.66 | **84.92** | **34.10** | **79.16** | **79.38** | **50.51** | **66.12** |
| LLaMA-3-8B | Dense | 60.19 | 72.77 | 81.35 | 34.80 | 79.71 | 80.09 | 50.43 | 65.62 |
| | SparseGPT | 34.26 | 56.75 | 66.51 | 16.80 | 63.28 | 42.09 | 21.42 | 43.02 |
| | w. LoRA | 44.93 | 62.35 | 61.59 | 21.80 | 70.57 | 60.06 | 31.23 | 50.36 |
| | **w. LoSA** | **46.09** | **62.98** | **62.87** | **23.80** | **72.09** | **62.96** | **33.11** | **51.99** |
| | Wanda | 27.36 | 49.96 | 53.33 | 12.00 | 56.04 | 31.86 | 17.41 | 35.42 |
| | w. LoRA | 39.52 | 57.22 | 61.92 | 17.40 | 68.44 | 56.40 | 28.24 | 47.02 |
| | **w. LoSA** | **43.12** | **61.01** | **62.61** | **23.20** | **70.95** | **60.35** | **31.40** | **50.37** |
| LLaMA-3.1-8B | Dense | 59.98 | 73.32 | 82.05 | 33.20 | 79.98 | 81.57 | 51.45 | 65.93 |
| | SparseGPT | 33.97 | 54.70 | 67.34 | 14.80 | 61.92 | 45.33 | 21.76 | 42.83 |
| | w. LoRA | 45.44 | 61.33 | 71.74 | 21.80 | 71.60 | 60.73 | 33.64 | 52.33 |
| | **w. LoSA** | **46.32** | **64.48** | **74.19** | **25.20** | **71.82** | **62.04** | **34.34** | **54.06** |
| | Wanda | 27.43 | 48.70 | 57.71 | 13.60 | 55.01 | 31.86 | 18.43 | 36.10 |
| | w. LoRA | 39.75 | 56.67 | 64.51 | 19.40 | 68.99 | 55.72 | 27.65 | 47.52 |
| | **w. LoSA** | **42.12** | **58.88** | **65.34** | **21.20** | **72.31** | **61.53** | **30.97** | **50.33** |

Table 25: Zero-shot accuracy results of `LoSA` for sparse Vicuna/OPT at 50/60/70% sparsity.

| Model | Method | HellaSwag | Winogrande | BoolQ | OBQA | PIQA | ARC-e | ARC-c | Mean |
|---|---|---|---|---|---|---|---|---|---|
| Vicuna-13B(50%) | Dense | 59.64 | 71.59 | 85.26 | 36.80 | 79.00 | 78.66 | 47.78 | 65.53 |
| | SparseGPT | 56.88 | 69.82 | 83.58 | 36.00 | 77.22 | 75.08 | 45.71 | 63.40 |
| | w. LoRA | 57.00 | 70.09 | 84.10 | 36.20 | 78.13 | 75.05 | 44.37 | 63.56 |
| | **w. LoSA** | **57.39** | **72.22** | **84.77** | **36.30** | **78.45** | **75.38** | **45.39** | **64.28** |
| | Wanda | 56.67 | 70.96 | 83.68 | 36.00 | 77.69 | 75.55 | 45.65 | 63.74 |
| | w. LoRA | 57.30 | 71.51 | 83.67 | 36.20 | 78.45 | 76.18 | 45.82 | 64.16 |
| | **w. LoSA** | **57.34** | **71.88** | **83.82** | **36.60** | **78.67** | **76.72** | **45.85** | **64.41** |
| Vicuna-13B(60%) | Dense | 59.64 | 71.59 | 85.26 | 36.80 | 79.00 | 78.66 | 47.78 | 65.53 |
| | SparseGPT | 51.75 | 68.27 | 81.09 | 32.20 | 74.80 | 71.65 | 42.02 | 60.25 |
| | w. LoRA | 54.23 | 69.77 | 81.56 | 32.30 | 77.15 | 71.21 | 42.61 | 61.26 |
| | **w. LoSA** | **54.67** | **70.09** | **81.62** | **32.40** | **77.31** | **72.90** | **42.71** | **61.67** |
| | Wanda | 51.46 | 68.43 | 81.71 | 31.60 | 74.65 | 71.84 | 41.64 | 60.19 |
| | w. LoRA | 54.08 | 68.55 | 80.31 | 32.00 | 76.93 | 71.91 | 42.41 | 60.89 |
| | **w. LoSA** | **54.85** | **69.14** | **81.28** | **32.80** | **77.20** | **71.97** | **42.66** | **61.42** |
| Vicuna-13B(70%) | Dense | 59.64 | 71.59 | 85.26 | 36.80 | 79.00 | 78.66 | 47.78 | 65.53 |
| | SparseGPT | 38.52 | 61.01 | 73.30 | 17.80 | 67.30 | 53.91 | 27.90 | 48.53 |
| | w. LoRA | 49.00 | 64.80 | 73.91 | 24.40 | 73.61 | 65.74 | 34.56 | 55.15 |
| | **w. LoSA** | **49.40** | **64.96** | **76.85** | **26.40** | **74.54** | **66.58** | **36.09** | **56.40** |
| | Wanda | 31.84 | 54.70 | 62.78 | 16.40 | 61.75 | 44.87 | 22.10 | 42.06 |
| | w. LoRA | 45.77 | 60.93 | 68.32 | 24.20 | 71.87 | 62.37 | 32.94 | 52.34 |
| | **w. LoSA** | **49.03** | **65.35** | **76.12** | **27.00** | **73.94** | **65.40** | **37.71** | **56.37** |
| OPT-13B(70%) | Dense | 52.43 | 65.04 | 65.93 | 27.20 | 75.84 | 67.13 | 32.94 | 55.22 |
| | SparseGPT | 40.94 | 61.40 | 63.65 | 21.00 | 69.10 | 52.44 | 25.94 | 47.68 |
| | w. LoRA | 46.79 | 60.77 | 63.79 | 26.40 | 72.96 | 59.55 | 29.10 | 51.34 |
| | **w. LoSA** | **46.83** | **61.01** | **68.65** | **27.00** | **72.99** | **60.02** | **29.69** | **52.31** |
| | Wanda | 34.36 | 55.09 | 55.02 | 15.60 | 62.89 | 43.73 | 23.89 | 41.51 |
| | w. LoRA | 44.81 | 59.69 | 60.52 | 22.60 | 71.93 | 55.85 | 28.67 | 49.16 |
| | **w. LoSA** | **45.20** | **59.91** | **60.96** | **24.80** | **73.39** | **57.65** | **29.01** | **50.13** |

Table 26: Zero-shot accuracy results of `LoSA` for sparse LLaMA-2-7B at N:M sparsity.

| Sparsity | Method | HellaSwag | Winogrande | BoolQ | OBQA | PIQA | ARC-e | ARC-c | Mean |
|---|---|---|---|---|---|---|---|---|---|
| 2:8 | Dense | 57.17 | 68.90 | 77.74 | 31.40 | 78.07 | 76.39 | 43.52 | 61.88 |
| | SparseGPT | 27.82 | 47.99 | 43.67 | 13.20 | 54.62 | 28.62 | 16.98 | 33.27 |
| | w. LoRA | 33.26 | 52.09 | 62.20 | 19.00 | 61.21 | 42.13 | 20.31 | 41.46 |
| | **w. LoSA** | **36.14** | **54.85** | **62.32** | **19.20** | **63.44** | **47.72** | **22.70** | **43.76** |
| | Wanda | 26.19 | 50.83 | 37.83 | 13.80 | 52.88 | 26.52 | 20.90 | 32.71 |
| | w. LoRA | 29.10 | 51.54 | 62.20 | 14.40 | 58.22 | 37.33 | 18.26 | 38.72 |
| | **w. LoSA** | **33.24** | **54.30** | **62.28** | **17.00** | **61.64** | **43.06** | **21.84** | **41.91** |

