# OpenReview forum: "Dynamic Low-Rank Sparse Adaptation for Large Language Models"
_ICLR.cc/2025/Conference — ICLR 2025 Poster_

### Official Review · Reviewer_VoNN · 2024-10-31

**Soundness:** 3
**Presentation:** 2
**Contribution:** 2
**Rating:** 3
**Confidence:** 4

**Summary:**

This paper introduces dynamic Low-rank Sparse Adaptation (LoSA), a method that integrates low-rank adaptation into LLM sparsity within a unified framework, thereby enhancing the performance of sparse LLMs without increasing the inference latency. In particular, LoSA dynamically sparsifies the LoRA outcomes based on the corresponding sparse weights during fine-tuning, thus guaranteeing that the LoRA module can be integrated into the sparse LLMs post-training. Besides, LoSA leverages Representation Mutual Information (RMI) as
an indicator to determine the importance of layers, thereby efficiently determining the layer-wise sparsity rates during fine-tuning. Predicated on this, LoSA adjusts the rank of the LoRA module based on the variability in layer-wise reconstruction errors, allocating an appropriate fine-tuning for each layer to reduce the output discrepancies between dense and sparse LLMs.

**Strengths:**

Extensive experiments tell that LoSA can efficiently boost the efficacy of sparse LLMs within a few hours, without introducing any additional inferential burden. For example, LoSA reduced the perplexity of sparse LLaMA-2-7B by 68.73 and increased zero-shot accuracy by
16.32%, achieving a 2.60× speedup on CPU and 2.23× speedup on GPU.

**Weaknesses:**

The novelty may be limited. This paper proposes three improvements including dynamic sparsity rates across layers with RMI, dynamic rank allocation, and progressive pruning. Theses ideas have been known to be effective for pruning. For example, [A1] atomically determines the   layer-wise sparsity ratio and [A2,A3] uses MI (and HSIC) in model pruning.  [A4,A5] investigates rank allocation. Progressive pruning has been commonly used in pruning and proved to be effective such as [A6,A7]. It mentions to merge sparse lora with sparse weights for actual acceleration. This idea is straightforward and [A7] already investigates sparse lora with sparse weights so that they can be merged. The adoption of RMI generally follows [Zheng et al., An Information Theory-inspired Strategy for
Automatic Network Pruning, 2021]. It seems to change the CNN model to LLMs with almost the same RMI method in [Zheng et al, 2021].  The technical contribution may be limited. It is better to highlight the unique contributions.


The baseline is not enough. The proposed method is a finetuning method. SparseGPT and wanda are PTQ methods without finetuning. It may not be fair to compare with  SparseGPT or wanda. The actual baseline only has lora. It may be better to compare with other finetuning based pruning methods for LLMs such as [A7], LLM-pruner or SliceGPT. Although most methods are structured pruning, it is easy to adapt the proposed method for structured pruning with such as wanda-sp of structured mask. It adopts multiple improvements and it is not surprising to be better than the naïve lora. It may be better to compare with related works focusing on pruning and finetuning LLMs.

The setting with  2:8 or mixed 2:8 sparsity may not be solid. The GPU compiler can accelerate a specific N:M sparsity such as 2:4. It may not be able to accelerate 2:8 or mixed 2:8 sparsity. The experiments for this part only demonstrate the accuracy performance. But the actual acceleration may be limited without the support from compiler.  It may be better to discuss  this setting.


The lora in the baseline finetunes the whole model so that the model is not sparse if merged. It may be a bit strange to compare between a dense model with lora and a sparse model with losa. It may be better to provide a new baseline such as SparseGPT + sparse lora or wanda + sparse lora. The sparse lora can be merged with the sparse model. It should be a more direct baseline with uniform sparsity rates and the same ranks in the merged model, in contrast to the dynamic sparsity and dynamic ranks. It may be more reasonable to compare the speed of this baseline and the proposed method as they are both sparse.

Reference

[A1] Layer-adaptive sparsity for the Magnitude-based Pruning

[A2] Layer-wise Model Pruning based on Mutual Information

[A3] MPruner: Optimizing Neural Network Size with CKA-Based Mutual Information Pruning

[A4] AdaLoRA: Adaptive Budget Allocation for Parameter-Efficient Fine-Tuning

[A5] ALoRA: Allocating Low-Rank Adaptation for Fine-tuning Large Language Models

[A6] Fast and Effective Weight Update for Pruned Large Language Models

[A7] LoRAPrune: Structured Pruning Meets Low-Rank Parameter-Efficient Fine-Tuning

**Questions:**

see the weakness.

It is better to highlight the unique contributions.

 It may be better to compare with related works focusing on pruning and finetuning LLMs.

 It may be better to discuss  this setting.

The baseline can be improved.

---

> ### Author Response · Authors · 2024-11-21
> **Response to Reviewer VoNN (1/4)**
>
> **We appreciate the time and effort you put in reviewing our work, as well as your detailed comments and valuable questions. We understand the concerns you raised and we're pleased to address these concerns. We also appreciate the opportunity to provide some clarification.**
>
> > W1: The novelty may be limited.
>
> **Answer (1/2):** We appreciate the reviewer's feedback and would like to clarify the unique contributions of our work, in contrast to prior methods. Below, we highlight our novel contributions and compare them to the cited works:
>
> 1. **Comparison of A1-A3:** LAMP [A1] proposes an importance score for global pruning, which is a rescaled version of weight magnitude that incorporates the model-level $\ell_2$ distortion incurred by pruning. This approach is significantly different from our method, which determines layer-wise sparsity rates based on mutual information (MI). A2 proposes retaining neurons in each layer that have high MI values with the neurons retained in the previous layer. This pruning is performed top-down, starting from the softmax layer and progressing to the embedding layer. However, pruning the 1.1B BERT-Extra-Large with A2 requires 16 V100 32GB GPUs, making it impractical for larger models ranging from 7B to 70B parameters. MPruner [A3] uses MI to evaluate inter-layer similarity and clusters layers based on their similarity to identify and remove redundant layers. However, the coarse-grained approach of removing entire layers results in a significant accuracy drop for LLMs, particularly under high pruning rates. Although LoSA, A2, and A3 all utilize MI to determine inter-layer redundancy, LoSA focuses on using MI to determine the layer-wise sparsity rates for sparse LLMs. This is an issue that neither A2 nor A3 addresses and represents a key distinction between LoSA and these approaches.
> 2. **Comparison of A4-A5:** LoSA, AdaLoRA [A4], and ALoRA [A5] all focus on developing efficient methods for rank allocation. However, there are significant methodological differences among them. AdaLoRA parameterizes incremental updates using singular value decomposition (SVD) and dynamically adjusts the rank of the incremental matrices based on a designed importance metric. ALoRA evaluates the importance of each rank, iteratively prunes low-contribution ranks, and reallocates resources to achieve dynamic adjustment of ranks. LoSA is a method specifically designed for sparse LLMs, dynamically adjusting parameters by allocating ranks based on the reconstruction error of sparse LLMs. Allocating ranks based on reconstruction error is a novel rank allocation approach proposed in our paper.
> 3. **Comparison of A6-A7:** Although our method also uses progressive pruning like [A6, A7], the insight behind our method is that due to the large number of LLMs parameters, pruning all weights at once will cause the reconstruction error of sparse LLMs to increase abnormally, making it difficult to restore the accuracy of sparse LLMs through fine-tuning. Therefore, we extend the one-shot pruning to progressive pruning. This gradual application of pruning effectively mitigates the output deviation caused by one-shot pruning. At the same time, in the process of progressive pruning, we also increase or decrease the rank of low-rank adaptation according to the changes in the reconstruction error of the sparse model, dynamically adjust the fine-tuning parameter allocation to minimize the reconstruction error of sparse LLMs. This approach of using both progressive pruning and dynamic rank allows us to more effectively fine-tune sparse LLMs to improve their performance, which is the unique contribution of our method.
> 4. **Comparison of A7:** Although our approach allows the integration of LoRA weights with sparse LLMs like LoRAPrune [A7], it is important to note a key difference: LoRAPrune [A7] is designed for structural pruned LLMs, where LoRA and pruned weights can be seamlessly merged by simply adjusting the input and output dimensions of LoRA. However, this method is not compatible with unstructured pruning, and LoRAPrune cannot be directly extended to fine-tuning unstructured pruned models. We propose simultaneously sparsifying both the low-rank adaptation and the LLMs to preserve the original acceleration advantages of sparse LLMs. Our method can integrate with any training-free unstructured pruning approach. Given the generality and versatility of our approach, we believe that LoSA makes a unique and valuable contribution to the community.

---

> ### Author Response · Authors · 2024-11-21
> **Response to Reviewer VoNN (2/4)**
>
> **Answer (2/2):**
>
> 5. **Comparison of Zheng et al. (2021):** It is important to emphasize that using RMI to determine the layer-wise sparsity rates of LLMs is just one subset of our numerous innovations. Additionally, while we acknowledge that our work is inspired by the methodology presented in Zheng et al. (2021), we believe that our paper introduces significant novel contributions by extending and adapting the RMI from CNNs to LLMs. Zheng et al. (2021) utilized mutual information to determine the layer-wise pruning rates for CNNs, followed by fine-tuning the one-shot pruned CNNs. This work only demonstrated the feasibility of using mutual information as a metric for pruning CNNs. However, due to the significant structural differences between LLMs and CNNs, the application of mutual information for pruning LLMs has not yet been explored. In contrast, LoSA employs a progressive sparse fine-tuning approach to obtain the final sparse LLMs. During this process, the relative importance of each layer in the LLMs changes due to weight updates during fine-tuning. However, given the massive number of parameters in LLMs, determining the relative importance of each layer is both challenging and time-consuming. In this paper, we derive the use of the RMI metric to efficiently and rapidly determine the relative importance of each layer in LLMs during sparse fine-tuning. Through extensive experiments, we validate the effectiveness of the RMI metric in determining sparsity rates for LLMs. Our method is the first to leverage mutual information theory to guide the pruning of LLMs, providing valuable insights to the research community. We acknowledge that the explanation of these unique contributions could be made more explicit in the paper. We will revise the manuscript to better highlight how our work differs from and advances beyond Zheng et al. (2021), emphasizing the innovations specific to LLMs.
> 6. **Contributions of our method:** our LoSA method addresses multiple challenges in a single algorithm framework, including determining the sparsity rates of the LLMs, allocating fine-tuning parameters, and resolving the issue of incompatibility between low-rank adaptation weights and sparse LLMs. By providing an end-to-end solution, LoSA effectively tackles various challenges in fine-tuning sparse LLMs. It significantly improves the accuracy of existing sparse LLMs and achieves faster inference compared to LoRA on both CPUs and GPUs. This substantially enhances the deployability of sparse LLMs across different hardware platforms, making a great contribution to the community.
>
> We hope this clarification highlights the uniqueness and impact of LoSA, and we are open to further suggestions to improve the presentation of our contributions. Thank you for your valuable feedback!

---

> ### Author Response · Authors · 2024-11-21
> **Response to Reviewer VoNN (3/4)**
>
> > W2: The baseline is not enough.
>
> Thank you for your valuable suggestions!
> 1. Firstly, it should be noted that our paper focuses on fine-tuning unstructured pruned LLMs. SparseGPT and Wanda are popular unstructured pruning methods for LLMs, and LoSA is designed to fine-tune unstructured pruned LLMs obtained by SparseGPT and Wanda. Our paper demonstrates the performance improvements that LoSA brings to these methods. Additionally, LoRA is a highly effective fine-tuning approach and a widely adopted method for fine-tuning unstructured pruned LLMs (the Wanda paper also shows results using LoRA to fine-tune unstructured pruned LLMs). Therefore, we believe that comparing our LoSA method with the LoRA baseline is reasonable and enough.
> 2. Secondly, based on your suggestions, we extended the LoSA method to fine-tune structured pruned LLMs and used the wanda-sp method for pruning. We compared LoSA with SliceGPT, LLM-Pruner, and LoRAPrune on the LLaMA-7B model with 20% pruning rate. The experimental results are as follows:
>
> |Method|WikiText2 PPL|BoolQ|PIQA|HellaSwag|WinoGrande|ARC-e|ARC-c|OBQA|Average|
> |-|-|-|-|-|-|-|-|-|-|
> |Dense|5.69|73.18|78.35|72.99|67.01|67.45|41.38|42.40|63.25|
> |SliceGPT|8.71|37.89|64.09|45.67|62.75|53.62|31.74|33.20|46.99|
> |LLM-pruner|8.14|69.54|76.44|68.11|65.11|63.43|37.88|40.00|60.07|
> |LoRAPrune|7.63|65.82|79.31|70.00|62.76|65.87|37.69|39.14|60.05|
> |**LoSA**|**7.07**|**71.67**|78.17|**71.56**|**65.86**|**66.93**|**40.66**|**40.50**|**62.19**|
>
> Our LoSA method outperforms SliceGPT, LLM-Pruner, and LoRAPrune as shown in the table, demonstrating the superior performance of LoSA. We have also included the experimental data on fine-tuning structured pruned LLMs in the revised version. Thank you again for your valuable suggestions.

---

> ### Author Response · Authors · 2024-11-21
> **Response to Reviewer VoNN (4/4)**
>
> > W3: The setting with 2:8 or mixed 2:8 sparsity may not be solid.
>
> Thank you for your valuable suggestions!
> 1. Since mixed 2:8 sparsity is a specific type of unstructured sparsity pattern, it can also leverage the nm-vllm sparse inference engine to achieve accelerated inference on GPUs. The acceleration effect is as follows:
>
> |Speed|Dense|Mixed 2:8|
> |-|-|-|
> |Throughput (tokens/s)|57.35|133.40|
> |Speedup|1.00$\times$|2.33$\times$|
>
> 2. Secondly, a major contribution of our paper is determining the layer-wise sparsity rates for LLMs. For N:M sparsity, we use our RMI-based sparsity rates method to determine the N value in mixed N:M sparsity. However, for 2:4 sparsity, our RMI-based sparsity rates method does not work.
> 3. Additionally, based on your suggestions, we also conducted experiments on 2:4 sparsity and mixed 2:4 sparsity on LLaMA2-7B obtained by Wanda method. For 2:4 sparsity, we do not use the RMI-based sparsity rates method. In contrast, for mixed 2:4 sparsity, we apply the RMI-based sparsity rates method. The experimental results are as follows.
>
> |Method|Sparsity|WikiText2 PPL|HellaSwag|WinoGrande|BoolQ|OBQA|PIQA|ARC-e|ARC-c|Average|
> |-|-|-|-|-|-|-|-|-|-|-|
> |Dense|0%|5.12|57.17|68.90|77.74|31.40|78.07|76.39|43.52|61.88|
> |Wanda|2:4 sparsity|11.02|40.92|62.43|67.65|24.20|70.84|61.78|31.20|51.29|
> |w. LoRA|2:4 sparsity|8.27|50.37|64.80|72.81|27.60|75.19|69.40|35.58|56.54|
> |w. LoSA|2:4 sparsity|7.91|51.24|65.79|73.97|29.00|76.17|70.76|36.62|57.65|
> |**w. LoSA**|**Mixed 2:4 sparsity**|**7.72**|**51.85**|**66.01**|**74.51**|**29.70**|**76.54**|**71.08**|**37.26**|**58.14**|
>
> LoSA improves accuracy for both 2:4 sparsity and mixed 2:4 sparsity and outperforms LoRA. We also present experimental data showing the use of the nm-vllm sparse inference engine to accelerate inference on GPUs for 2:4 sparsity and mixed 2:4 sparsity models. Since both 2:4 sparsity and Mixed 2:4 sparsity have a sparsity rate of 50%, the acceleration effects are essentially the same.
>
> |Speed|Dense|2:4|Mixed 2:4|
> |-|-|-|-|
> |Throughput (tokens/s)|57.35|98.23|98.35|
> |Speedup|1.00$\times$|1.71$\times$|1.71$\times$|
>
> > W4: Comparison with Sparse LoRA
>
> Thank you for your suggestions. We present the comparison of accuracy and inference acceleration between Sparse LoRA, LoRA and LoSA. Specifically, we show the results for the 70% sparse LLaMA-2-7B model.
>
> Accuracy comparison:
>
> |Method|WikiText2 PPL|HellaSwag|WinoGrande|BoolQ|OBQA|PIQA|ARC-e|ARC-c|Average|
> |-|-|-|-|-|-|-|-|-|-|
> |Dense|5.12|57.17|68.90|77.74|31.40|78.07|76.39|43.52|61.88|
> |SparseGPT|27.42|33.08|58.41|64.89|17.40|62.46|43.22|22.01|43.07|
> |w. Sparse LoRA|11.26|43.63|62.06|63.46|22.80|70.84|57.22|29.01|49.86|
> |w. LoRA|11.06|44.80|62.90|63.36|24.20|71.22|58.71|30.12|50.76|
> |**w.LoSA**|**10.82**|**46.06**|**63.85**|**70.15**|**24.80**|**71.93**|**60.44**|**30.35**|**52.51**|
> |Wanda|79.67|27.92|49.33|52.87|12.60|55.33|30.60|18.69|35.33|
> |w. Sparse LoRA|12.74|40.53|56.84|64.08|22.20|68.53|55.77|26.37|47.76|
> |w. LoRA|12.57|40.77|57.22|64.19|22.40|68.55|57.32|26.79|48.18|
> |**w.LoSA**|**10.94**|**45.10**|**60.93**|**67.65**|**25.20**|**71.06**|**62.50**|**29.10**|**51.65**|
>
> Inference acceleration comparison:
>
> |70% Sparsity|Dense|LoRA|Sparse LoRA|LoSA|
> |-|-|-|-|-|
> |Throughput (tokens/s)|57.35|98.10|127.63|127.69|
> |Speedup|1.00$\times$|1.71$\times$|2.23$\times$|2.23$\times$|
>
> Since Sparse LoRA's Low-rank adaptation is also sparse, its accuracy is worse than LoRA, and it is also much worse than LoSA. Additionally, since both Sparse LoRA and LoSA can be merged into sparse LLMs, their inference acceleration effects are basically the same, and both outperform LoRA. Overall, our proposed LoSA method outperforms both Sparse LoRA and LoRA in terms of accuracy and inference acceleration.
>
> **Finally, I hope that my response has addressed your concerns, and I kindly ask for a more favorable evaluation of our work. Thank you!**

---

> ### Author Response · Authors · 2024-11-29
> **A kind reminder**
>
> Dear Reviewer VoNN,
>
> Thanks again for your valuable time and insightful comments. As the deadline for the Author/Reviewer discussion is approaching, it would be nice of you to let us know whether our answers have solved your concerns so that we can better improve our work. We are happy to provide any additional clarifications that you may need.
>
> Best regards,
>
> Authors

---

> ### Author Response · Authors · 2024-12-01
> **Last two day reminder**
>
> Dear Reviewer VoNN,
>
> We have put in many efforts to address the raised questions and concerns. As the ICLR rebuttal period is approaching its end, we kindly remind you to review our submitted response. Additionally, we sincerely hope that you can reconsider the score of our paper. Your feedback is essential for finalizing our work.
>
> Thank you for your attention.
>
> Best regards,
>
> Submission417 Authors

---

> ### Comment · Reviewer_VoNN · 2024-12-02
>
> Thanks for the rebuttal. After going through the rebuttal, I still have some concerns for the novelty. As I mentioned in my review,  this paper proposes three improvements including dynamic sparsity rates across layers with RMI, dynamic rank allocation, and progressive pruning, which have been explored and demonstrated effect in LLM pruning. I mentioned some related works and I know they are different from this paper. I list these papers to show that the ideas in this paper have been explored in pruning of DL or LLM models.
>
> Furthermore, some technical components in the paper are almost the same as the previous works. The RMI method in this paper is almost the same as that in [Zheng et al. (2021)]. The problem formulation, algorithm design and RMI definition based on HSIC are exactly the same. The authors may change the expressions but they are indeed the same. The progressive pruning method of this paper is also the same as the progressive sparsity in [A6] for LLM pruning.
>
> The rebuttal highlights other things. [Zheng et al. (2021)] uses RMI to determine the layer-wise sparsity for convolutional layers, and this paper uses RMI to determine the layer-wise sparsity for linear layers. I do not see there are too many differences here. Convolutional layers are specific forms of linear layers, and this step just determines the sparsity rates. How to prune under the given sparsity rates are determined by other methods suitable for CNN or LLM. It does not seem to be a significant contribution to determine the sparsity rates for linear layers compared with convolutional layers using the same method. The RMI method relies on the output of the layers, without touching the unique computation patterns of linear or convolutional layers.

---

> > ### Author Response · Authors · 2024-12-03
> > **Response to Reviewer VoNN**
> >
> > Thank you for your further response.
> >
> > 1. We acknowledge that the three ideas proposed in this paper have been explored in pruning of DL or LLM models, and these ideas have indeed been shown to be effective for LLMs pruning, but as you pointed out, **"I mentioned some related works and I know they are different from this paper."** We believe that the novelty of our method lies in the **technical novelty**, which is reflected in the fact that our method proposes **new solutions to the original problems**, including a new sparsity rate determination method based on RMI, a rank assignment scheme based on reconstruction error, and dynamic adjustment of ranks during progressive pruning. These unique technical contributions emphasize the novelty of our method.
> >
> > 2. We acknowledge that the sparsity rate determination method based on RMI is inspired by [Zheng et al. (2021)], however, **the core contribution is that we extend RMI to LLM, integrate it with dynamic rank assignment, and incorporate it into the progressive pruning framework. These components together form a unified and automated framework**, which is not available in previous work. And we agree that the RMI method relies on the output of the layers, without touching the unique computation patterns of linear or convolutional layers. However, we need to point out that due to the huge difference in the network structure of CNN and LLM, and one is used for visual tasks and the other is used for language tasks, there are significant differences between the feature maps of the two. Although feature map-based RMI has been successful in determining the sparsity rate of CNN, in this paper, we derive a method to use the RMI metric to effectively and quickly determine the relative importance of each layer in LLM during sparse fine-tuning. Through extensive experiments, we verify the effectiveness of the RMI metric in determining the sparsity rate of LLM. **Our method is the first method to use mutual information theory to guide LLM pruning, providing valuable insights to the community.**
> >
> > 3. We acknowledge that the progressive pruning method in this paper is the same as [A6], but it should be pointed out that we and [A6] both adopt the cubic sparsity schedule proposed in [Zhu & Gupta, 2017, To prune, or not to prune: exploring the efficacy of pruning for model compression]. We mentioned this in our paper, "We determine the progressive sparsity rate using the cubic sparsity schedule proposed by (Zhu & Gupta, 2017) (Line 277)", and [A6] also mentioned this in Section 3.2, "We adopt cubic sparsity schedule from (Zhu & Gupta, 2017)". In addition, although we and [A6] use the same progressive sparsity method, **we dynamically adjust the rank during the progressive sparsification process** (Eq. 9 in our paper). This method of dynamically adjusting the sparsity rate and rank is completely different from previous works.
> >
> > We hope that the above response has addressed your concerns and look forward to your further response.

---

### Official Review · Reviewer_7hsM · 2024-11-02

**Soundness:** 1
**Presentation:** 2
**Contribution:** 2
**Rating:** 3
**Confidence:** 4

**Summary:**

This paper presents an approach called "Dynamic Low-Rank Sparse Adaptation" (LoSA) that seeks to address the challenges of performance degradation associated with sparsifying LLMs. The paper introduces a unified framework that combines low-rank adaptation with sparsity, aiming to improve efficiency while maintaining model performance.

**Strengths:**

1. The paper offers extensive experimental results across various models (e.g., LLaMA-2, Vicuna, OPT) and datasets.

2. The authors provide an explanation of their methodology, from dynamic sparsification to adaptive rank allocation.

**Weaknesses:**

1. This paper ignores previous works [1-4] on sparse PEFT using Low-Rank Adaptation. I did not see any comparisons or discussions with previous similar works. It's extremely hard to tell if this work has enough contributions to the area since the author choose such narrow baselines (SparseGPT, Wanda).

2. Without through discussions with previous similar works, it's hard to judge the novelty of this work.

3. The author could also compare with some latest works of LLM post-training pruning works.

4. The improvements over baselines are incremental, some of them are very marginal.

Given the current state of this paper, it's a clear reject for me.

[1] Sparse Low-rank Adaptation of Pre-trained Language Models, EMNLP'23

[2] LoRAPrune: Structured Pruning Meets Low-Rank Parameter-Efficient Fine-Tuning, ACL'24

[3] LoRAShear: Efficient Large Language Model Structured Pruning and Knowledge Recovery

[4] LoSparse: Structured Compression of Large Language Models based on Low-Rank and Sparse Approximation

**Questions:**

Please check the weakness. Please provide comprehensive experimental results and discussions.

---

> ### Author Response · Authors · 2024-11-21
> **Response to Reviewer 7hsM (1/2)**
>
> **We greatly appreciate your time, detailed comments, and valuable suggestions. We are pleased to address your concerns and make some clarifications.**
>
> > W1 & W2: This paper ignores previous works [1-4] on sparse PEFT using Low-Rank Adaptation.
>
> Thank you for your valuable suggestions!
> 1. Firstly, it should be noted that our paper focuses on fine-tuning unstructured pruned LLMs. SparseGPT and Wanda are popular unstructured pruning methods for LLMs. Our paper demonstrates the performance improvements that LoSA brings to these methods. Additionally, LoRA is a highly effective fine-tuning approach and a widely adopted method for fine-tuning unstructured pruned LLMs (the Wanda paper also shows results using LoRA to fine-tune unstructured pruned LLMs). Therefore, We compare our LoSA with LoRA.
> 2. Secondly, SoRA [1] is to dynamically adjust the rank of low-rank adaptation in the training process with a sparse gating unit trained by proximal gradient method. The only similarity between LoSA and SoRA lies in their adaptive rank adjustment in low-rank adaptation during fine-tuning. However, there are significant differences between the two methods. LoSA dynamically allocates ranks to each layer based on reconstruction error. Moreover, LoSA can be merged into sparse LLMs, whereas SoRA cannot. Additionally, LoSA also introduces a method for determining layer-wise sparsity rates for LLMs based on mutual information and adopts an iterative sparse fine-tuning approach. In the revised version, we have added a discussion of SoRA in the related work section. Based on your suggestion, we compared the SoRA and LoSA methods:
>
> |Method|WikiText2 PPL|HellaSwag|WinoGrande|BoolQ|OBQA|PIQA|ARC-e|ARC-c|Average|
> |-|-|-|-|-|-|-|-|-|-|
> |Dense|5.12|57.17|68.90|77.74|31.40|78.07|76.39|43.52|61.88|
> |Wanda|79.67|27.92|49.33|52.87|12.60|55.33|30.60|18.69|35.33|
> |SoRA|11.89|41.37|57.87|64.95|23.40|68.78|58.25|27.17|48.83|
> |**LoSA**|**10.94**|**45.10**|**60.93**|**67.65**|**25.20**|**71.06**|**62.50**|**29.10**|**51.65**|
>
> The results show that LoSA method outperforms SoRA. This is evident because SoRA only dynamically adjust the rank. In contrast, LoSA dynamically adjusts the rank based on reconstruction error, determines layer-wise sparsity rates for sparse LLMs, and adopts iterative sparse fine-tuning. These strategies ensure that LoSA achieves better accuracy than SoRA.
>
> 3. Thirdly, both LoRAPrune [2] and LoRAShear [3] fine-tune structured pruned LLMs. We have discussed related content in the related work section. In these works, low-rank adaptation is dense and cannot be merged into unstructured pruned LLMs. Consequently, this paper aims to explore effective techniques for integrating low-rank adaptations into the unstructured pruned LLMs. The goal is to ensure that sparse LLMs and low-rank adaptations share the same sparse mask, thereby the model’s sparsity is preserved and inference latency remains unaffected. Since our paper focuses on fine-tuning unstructured pruned LLMs, we do not include them in the comparison baselines. According to your suggestion, we extended the LoSA method to fine-tune the structured pruned LLMs, and used the wanda-sp method to obtain the LLaMA-7B with 20% pruning rate. We compared the LoRAPrune and LoRAShear. The experimental data of LoRAPrune and LoRAShear come from LoRAShear paper.
>
> |Method|BoolQ|PIQA|HellaSwag|WinoGrande|ARC-e|ARC-c|OBQA|Average|
> |-|-|-|-|-|-|-|-|-|
> |Dense|73.18|78.35|72.99|67.01|67.45|41.38|42.40|63.25|
> |LoRAPrune|65.82|**79.31**|70.00|62.76|65.87|37.69|39.14|60.05|
> |LoRAShear|70.17|76.89|68.69|65.83|64.11|38.77|39.97|60.63|
> |**LoSA**|**71.67**|78.17|**71.56**|**65.86**|**66.93**|**40.66**|**40.50**|**62.19**|
>
> LoSA outperforms LoRAPrune and LoRAShear.
>
> 4. Finally, LoSparse [4] performs structured pruning on smaller models like BERT, using a combination of low-rank and sparse matrices to approximate the original weight matrix. LoSparse involves structured pruning of the sparse matrix, followed by training the low-rank matrix to compensate for the sparse matrix, resulting in structured pruned BERT. The only similarity between LoSA and LoSparse lies in their use of iterative pruning. However, there are significant differences between two methods. LoSA introduces innovations such as sparsity alignment of LoRA and sparse LLMs weights, determination of layer-wise sparsity rates for LLMs, and adaptive adjustment of the rank in LoRA during fine-tuning. In our revised version, we added a discussion on LoSparse in the related work section. Since the LoSparse method does not present experimental results on LLMs, and LoSparse requires the same memory as full model fine-tuning, we lack sufficient GPU resources to demonstrate LoSparse results on models larger than 7B. As a result, it is difficult to directly compare the advantages and disadvantages of LoSA and LoSparse.
>
> We have included all experimental data mentioned above in the revised version. Thank you again for your valuable suggestions.

---

> ### Author Response · Authors · 2024-11-21
> **Response to Reviewer 7hsM (2/2)**
>
> > W3: The author could also compare with some latest works of LLM post-training pruning works.
>
> Thank you for your suggestions. It should be noted that our LoSA method is designed for fine-tuning unstructured pruned LLMs, which means that LoSA can be combined with any LLM post-training unstructured pruning methods to enhance their accuracy. We have demonstrated the improvement that LoSA brings to SparseGPT and Wanda, both of which are popular post-training unstructured pruning methods for LLMs. In the table below, we further show the performance improvements of LoSA on the latest LLM post-training pruning methods, including Pruner-Zero [A1] and ALPS [A2]. All experimental data are based on a 70% sparse LLaMA2-7B model.
>
> |Method|WikiText2 PPL|HellaSwag|WinoGrande|BoolQ|OBQA|PIQA|ARC-e|ARC-c|Average|
> |-|-|-|-|-|-|-|-|-|-|
> |Dense|5.12|57.17|68.90|77.74|31.40|78.07|76.39|43.52|61.88|
> |Pruner-Zero|103.15|27.56|50.99|41.93|13.00|56.90|34.47|18.60|34.78|
> |w. LoRA|11.56|43.43|60.46|67.19|21.00|70.40|59.60|27.47|49.94|
> |**w. LoSA**|**10.78**|**45.56**|**62.10**|**69.15**|**25.00**|**71.73**|**61.08**|**29.45**|**52.01**|
> |ALPS|19.31|38.35|61.96|64.59|22.20|66.82|48.37|24.95|46.75|
> |w. LoRA|10.83|47.54|62.88|69.11|27.00|73.23|61.70|29.78|53.03|
> |**w. LoSA**|**10.28**|**49.90**|**64.34**|**71.38**|**28.10**|**75.24**|**63.78**|**31.27**|**54.86**|
>
> From the data in the table, we can observe that the accuracy of post-training pruned LLMs has significantly decreased compared to the dense model. LoSA effectively improves the accuracy of post-training pruned LLMs, outperforming LoRA. We have also included the above experimental data in the revised version. If you would like us to show the fine-tuning effects of LoSA on other LLM post-training pruning works, please let us know.
>
> [A1] Pruner-Zero: Evolving Symbolic Pruning Metric from scratch for Large Language Models. ICML 2024.
>
> [A2] ALPS: Improved Optimization for Highly Sparse One-Shot Pruning for Large Language Models. NeurIPS 2024.
>
> > W4: The improvements over baselines are incremental, some of them are very marginal.
>
> We thank you for your comment and appreciate the opportunity to clarify our contributions:
>
> 1. Our paper’s experiments cover multiple architectures (LLaMA, Vicuna, OPT) and sizes, presenting results across different sparsity ratios. Performance improvements in both perplexity (Table 1), zero-shot accuracy (Table 2) and sppedup (Table 9) underscore LoSA’s adaptability across models and sparsity ratios. Experimental results show that LoSA has a significant improvement over the two training-free sparsity methods including Wanda and SprseGPT. For example, LoSA reduced the perplexity of 70% sparse LLaMA-2-7B using Wanda method by 68.73 and increased zero-shot accuracy by 16.32%, achieving a 2.60$\times$ speedup on CPU and 2.23$\times$ speedup on GPU. In the above case, LoSA has 1.63 lower perplexity than LoRA, 3.47% higher zero-shot accuracy, 0.31$\times$ faster speedup on CPU, and 0.52$\times$ faster speedup on GPU. This is a considerable improvement.
> 2. Additionally, the accuracy degradation of sparse LLMs obtained by SparseGPT and Wanda method is relatively small compared to dense models at lower sparsity rates. Thus, there is limited room for improvement in model accuracy through fine-tuning at low sparsity rates. As a result, LoSA offers less improvement for LLMs with low sparsity rates than that with high sparsity rates. However, LoSA further reduces the accuracy gap between low sparsity LLMs and dense LLMs. For instance, the zero-shot accuracy of a 50% sparse LLaMA2-7B model obtained through SparseGPT shows a gap of 2.59% compared to the dense model, while LoSA reduces this gap to 0.56%, outperforming the 0.85% achieved by LoRA method. Furthermore, LoSA enables the creation of a lossless 60% sparse LLaMA2-70B model, achieving a 0.84% higher accuracy compared to the LoRA method. In contrast, a 60% sparse LLaMA2-70B model obtained by SparseGPT exhibits a 2.87% accuracy drop relative to the dense model. The precision-lossless sparse LLMs achieve inference acceleration on both CPU and GPU, which significantly enhances practical deployability and once again proves the superiority of our method.
>
> Thank you for your consideration, and we look forward to your feedback.
>
> **Finally, I hope that my response has addressed your concerns, and I kindly ask for a more favorable evaluation of our work. Thank you!**

---

> ### Author Response · Authors · 2024-11-29
> **A kind reminder**
>
> Dear Reviewer 7hsM,
>
> Thanks again for your valuable time and insightful comments. As the deadline for the Author/Reviewer discussion is approaching, it would be nice of you to let us know whether our answers have solved your concerns so that we can better improve our work. We are happy to provide any additional clarifications that you may need.
>
> Best regards,
>
> Authors

---

> ### Author Response · Authors · 2024-12-01
> **Last two day reminder**
>
> Dear Reviewer 7hsM,
>
> We have put in many efforts to address the raised questions and concerns. As the ICLR rebuttal period is approaching its end, we kindly remind you to review our submitted response. Additionally, we sincerely hope that you can reconsider the score of our paper. Your feedback is essential for finalizing our work.
>
> Thank you for your attention.
>
> Best regards,
>
> Submission417 Authors

---

> ### Author Response · Authors · 2024-12-03
> **Looking forward to your response**
>
> Dear Reviewer 7hsM,
>
> As the deadline of the discussion phase is approaching, we kindly ask you to review our response to see whether it can address your concerns. Following your suggestions, we have compared LoSA with related works [1-4], demonstrated the novelty of our approach, and show that LoSA improves on the latest LLM post-training pruning works.
>
> If our response has addressed your concerns, would you mind reconsidering your score?
>
> Thank you once again for your effort and valuable feedback.

---

### Official Review · Reviewer_MFoH · 2024-11-03

**Soundness:** 2
**Presentation:** 3
**Contribution:** 3
**Rating:** 6
**Confidence:** 4

**Summary:**

The paper proposes Dynamic Low-Rank Sparse Adaptation (LoSA). This framework enhances sparse Large Language Models (LLMs) by integrating low-rank adaptation (LoRA) into the sparsity framework with dynamically adjusted layer-wise sparsity rates and rank allocations. LoSA utilizes representation mutual information (RMI) to determine layer importance for sparsity and reconstruction errors to allocate ranks, which purportedly improves performance without increasing inference latency.  Experimental results demonstrate that LoSA achieves considerable gains in accuracy, perplexity, and inference efficiency across various architectures and sizes.

**Strengths:**

1.  LoSA introduces a combined dynamic sparsity and rank adjustment mechanism for fine-tuning sparse LLMs. Using RMI for layer-wise sparsity rate determination and reconstruction errors for rank allocation seems a reasonable approach for preserving model performance under sparse conditions. Moreover, trying to match the sparsity pattern of the adaptation path BA and the pre-trained weight W is novel.

2. Comprehensive Empirical Evaluation: The paper’s experiments cover multiple architectures (LLaMA, Vicuna, OPT) and sizes, presenting results across different sparsity ratios. Performance improvements in both perplexity (Table 1) and zero-shot accuracy (Table 2) underscore LoSA’s adaptability across models and contexts, although some improvements at lower sparsity rates are modest.

3. Inference Efficiency: LoSA reportedly achieves significant inference speedups, which is crucial for deploying LLMs in resource-constrained environments. The throughput gains shown in Table 9 demonstrate LoSA’s potential in reducing inference latency compared to other sparsification methods, although these gains are achieved using specific sparse inference engines.

**Weaknesses:**

1. The paper lacks comparisons with adaptive LoRA methods like AdaLoRA[1] and SoRA[2], which are critical for evaluating LoSA’s performance among recent dynamic rank approaches. Without these comparisons, LoSA’s relative advantage remains unclear.

2.  The optimization setup in Eq. 5 (Section 2.2) assigns higher sparsity rates to layers with higher importance, which contradicts standard practices that seek to preserve the most important layers. This questionable logic may weaken the model’s representational power and suggest a potential flaw in LoSA’s sparsity allocation strategy.

3. The paper does not clearly explain how LoSA ensures consistency in sparsity across LoRA weights (BA matrices) and model weights (W matrices). Given the critical need for alignment in the sparse structure, the mechanism by which LoSA achieves this alignment is unclear, especially in the context of using SparseGPT or Wanda. This missing detail may complicate LoSA’s practical applicability.

4. While some ablation studies are presented, further analysis is needed on the soundness and effectiveness of the RMI-based sparsity and reconstruction-based rank allocation across different architectures.

[1] Zhang, Qingru, et al. "AdaLoRA: Adaptive budget allocation for parameter-efficient fine-tuning." arXiv preprint arXiv:2303.10512 (2023).

[2] Ding, Ning, et al. "Sparse low-rank adaptation of pre-trained language models." arXiv preprint arXiv:2311.11696 (2023).

**Questions:**

1   - Could authors provide results comparing LoSA with adaptive LoRA baselines like AdaLoRA and SoRA? These comparisons would help clarify how LoSA performs relative to other dynamic sparsity approaches.

2- In Eq. 5, why is the sparsity rate set to increase with the layer importance score? This setup seems counterintuitive since it would sparsify important layers more, potentially impacting the model’s performance. Could you explain this choice and its implications?

3- How does LoSA ensure consistent sparsity patterns between the LoRA (BA) and LLM (W) weights? Does the method sparsify entire rows or columns to maintain this alignment, or is there another approach? Further detail would clarify how LoSA integrates with existing sparse methods like SparseGPT and Wanda.

4- It would be helpful if authors could also elaborate on the computational overhead introduced by the RMI and reconstruction error-based adaptations. Specifically, how does this additional computation scale with model size?

 5- Could authors also clarify the interpretability of RMI as a sparsity allocation metric? While the Information Bottleneck principle supports its use, it would be helpful to see additional justification or experiments demonstrating that RMI consistently aligns with real layer importance (e.g., gradient-based layer importance) across diverse architectures.

6- Given that LoSA’s speedups are measured using specific sparse inference engines (e.g., DeepSparse, nm-vllm), how generalizable are these results to other deployment environments?

---

> ### Author Response · Authors · 2024-11-21
> **Response to Reviewer MFoH (1/2)**
>
> **We appreciate the time and effort you put in reviewing our work, as well as your detailed comments and valuable questions. We understand the concerns you raised and we're pleased to address these concerns.**
>
> > W1&Q1: Could authors provide results comparing LoSA with adaptive LoRA baselines like AdaLoRA and SoRA?
>
> Thank you for your suggestions. We present comparison of LoSA with AdaLoRA [1] and SoRA [2] below. Since AdaLoRA and SoRA have only been experimented on smaller models (such as BERT) and there is no experimental data for LLMs (such as LLaMA), we use the open source codes of AdaLoRA and SoRA to fine-tune 70% sparse LLaMA2-7B obtained by the Wanda. Since the rank of LoRA is 8, according to the original paper, the initial rank for each incremental matrix in AdaLoRA is 12. The rank of the SoRA method is set to 8, and other hyperparameters are set according to the original paper. Other experimental settings follow those in Section 3.1 and are aligned with the settings of LoRA and LoSA.
>
> |Method|WikiText2 PPL|HellaSwag|WinoGrande|BoolQ|OBQA|PIQA|ARC-e|ARC-c|Average|
> |-|-|-|-|-|-|-|-|-|-|
> |Dense|5.12|57.17|68.90|77.74|31.40|78.07|76.39|43.52|61.88|
> |Wanda|79.67|27.92|49.33|52.87|12.60|55.33|30.60|18.69|35.33|
> |LoRA|12.57|40.77|57.22|64.19|22.40|68.55|57.32|26.79|48.18|
> |AdaLoRA|12.08|41.01|57.78|64.73|23.00|69.09|57.77|26.90|48.61|
> |SoRA|11.89|41.37|57.87|64.95|23.40|68.78|58.25|27.17|48.83|
> |**LoSA**|**10.94**|**45.10**|**60.93**|**67.65**|**25.20**|**71.06**|**62.50**|**29.10**|**51.65**|
>
> The results show that LoSA outperforms AdaLoRA and SoRA. This is evident because AdaLoRA and SoRA only dynamically adjust the rank, and the weights of AdaLoRA and SoRA cannot be merged into sparse LLMs. In contrast, LoSA dynamically adjusts the rank based on reconstruction error, determines layer-wise sparsity rates for sparse LLMs, and adopts dynamic sparse fine-tuning. Additionally, LoSA weights can be merged into sparse LLMs. These strategies ensure that LoSA achieves better accuracy than AdaLoRA and SoRA. We have also included the experimental data in the revised version.
>
> > W2&Q2: In Eq. 5, why is the sparsity rate set to increase with the layer importance score?
>
> We apologize for any confusion caused during your reading, and your understanding is correct. We assign lower sparsity rates to the more important layers. We define the sparsity rate $s_i$ as the ratio of zero elements to the total number of elements in each layer. Consequently, Equation (5) is updated to $\textcolor{red}\min_{\boldsymbol{s}}  \boldsymbol{p}^T\boldsymbol{s} \, s.t.  \frac{1}{n}\sum_{i=1}^{n}s_i = {\Theta}$, changing the original maximization objective into a minimization one. We have corrected the above errors in the revised version of the manuscript, and we sincerely apologize again for any inconvenience caused by these oversights.
>
> > W3&Q3: How does LoSA ensure consistent sparsity patterns between the LoRA (BA) and LLM (W) weights?
>
> This issue can be illustrated with reference to Figure 1 and Equation 1 from the paper, which provide an intuitive explanation. Specifically, post-training sparsification methods like SparseGPT and Wanda essentially aim to derive a mask $\boldsymbol{M}$ for the weights $\boldsymbol{W}$ of the LLM. In the original LoRA, the $\boldsymbol{BA}$ matrices are dense. Consequently, the expression $\boldsymbol{M}\odot  \boldsymbol{W} + \boldsymbol{BA}$ cannot achieve sparsity consistency. LoSA addresses this by also sparsifying the $\boldsymbol{BA}$ matrices, applying the same mask $\boldsymbol{M}$ as the LLM weights $\boldsymbol{W}$. This results in $\boldsymbol{M}\odot  (\boldsymbol{W} + \boldsymbol{BA})$, thereby achieving sparsity consistency between the LLM weights $\boldsymbol{W}$ and the LoRA $\boldsymbol{BA}$  matrices.
>
> > W4: Further analysis is needed on the soundness and effectiveness of the RMI-based sparsity and reconstruction-based rank allocation across different architectures.
>
> Thank you for your suggestions. We further conducted ablation experiments on the OPT model, which is a non-LLaMA architecture, using RMI-based sparsity and reconstruction-based rank allocation. Specifically, we used the Wanda method to obtain a 70% sparse OPT-13B. The results are as follows:
>
> |Method|WikiText2 PPL|HellaSwag|WinoGrande|BoolQ|OBQA|PIQA|ARC-e|ARC-c|Average|
> |-|-|-|-|-|-|-|-|-|-|
> |**LoSA**|**19.75**|**45.20**|**59.91**|**60.96**|**24.80**|**73.39**|**57.65**|**29.01**|**50.13**|
> |w/o  RMI-based sparsity|20.72|44.21|59.32|59.34|24.20|72.45|57.10|28.15|49.24|
> |w/o  rank allocation|20.55|44.84|59.66|59.24|24.40|72.69|57.07|28.41|49.47|
> |w/o  sparsity & allocation|21.48|43.35|58.78|58.65|23.90|72.09|56.45|27.56|48.68|
>
> We can see that removing either the RMI-based sparsity or the reconstruction-based rank allocation leads to a decrease in LoSA accuracy. The results demonstrate the soundness and effectiveness of the RMI-based sparsity and reconstruction-based rank allocation across different architectures.

---

> ### Author Response · Authors · 2024-11-21
> **Response to Reviewer MFoH (2/2)**
>
> > Q4: It would be helpful if authors could also elaborate on the computational overhead introduced by the RMI and reconstruction error-based adaptations. Specifically, how does this additional computation scale with model size?
>
> Thank you for your suggestion. We present the computation overhead for RMI-based sparsity and reconstruction error-based adaptations for the 7B to 70B models in the table below. All data is collected on a single NVIDIA A100 80GB GPU.
>
> |Computational overhead (seconds)|7B|8B|13B|30B|70B|
> |-|-|-|-|-|-|
> |RMI-based sparsity|48|34|74|140|332|
> |reconstruction error-based adaptations|46|32|71|132|321|
>
> We can observe that as the model size increases, the required computation time also increases. However, for the largest 70B model, the computation time for RMI-based sparsity and reconstruction error-based adaptations are only 332 seconds and 321 seconds, respectively, which are very fast and have minimal computational overhead.
>
> > Q5: Could authors also clarify the interpretability of RMI as a sparsity allocation metric? While the Information Bottleneck principle supports its use, it would be helpful to see additional justification or experiments demonstrating that RMI consistently aligns with real layer importance (e.g., gradient-based layer importance) across diverse architectures.
>
> Thank you for your suggestion. To the best of our knowledge, there is currently no work on gradient-based layer importance for LLMs, so we cannot compare the layer-wise sparsity rates obtained by the RMI metric with such approaches. If you have gradient-based layer importance methods for determining LLMs layer-wise sparsity rates, please let us know and we would be happy to add comparisons with these methods.
>
> We plot a line graph of layer-wise sparsity rates for different LLMs in Figure 3 of our paper. We observed that the RMI metric tends to assign lower sparsity rates to the initial and final layers of LLMs while allocating higher sparsity rates to the middle layers. This can be explained as follows: the initial layers of LLMs are typically responsible for processing low-level features of the input, which are crucial for the subsequent layers' understanding. The middle layers primarily handle information processing and feature extraction, where some redundancy often exists. The final layers focus on generating high-quality contextual representations, directly influencing the model's output. This sparsity allocation strategy is similar to the approach adopted by LLM-Pruner [1], which does not prune the first three layers or the last layer but prunes the middle layers. It also aligns with the findings of SLEB [2], which prunes LLMs by discarding unimportant layers. According to Table 4 in the SLEB paper, the discarded layers are mainly concentrated in the middle parts of the LLMs.
>
> [1] LLM-Pruner: On the Structural Pruning of Large Language Models. NeurIPS 2023.
>
> [2] SLEB: Streamlining LLMs through Redundancy Verification and Elimination of Transformer Blocks. ICML 2024.
>
> > Q6: Given that LoSA’s speedups are measured using specific sparse inference engines (e.g., DeepSparse, nm-vllm), how generalizable are these results to other deployment environments?
>
> Thank you for your questions.
>
> 1. First, due to the sparsity of weights in unstructured pruning, we must use a specific sparse inference engine to accelerate inference. We use DeepSparse and nm-vllm to accelerate inference on general deployment environments, including CPU and GPU.
> 2. For CPUs, the DeepSparse tool supports architectures including: x86 AVX2, AVX-512, AVX-512 VNNI, and ARM v8.2+, which covers most Intel, AMD, and Apple M-series CPUs.
> 3. For GPUs, as long as the GPU supports CUDA installation, inference acceleration can be achieved using the nm-vllm tool. Similarly, if CUDA installation is supported for other deployment environments, nm-vllm can also be used to achieve acceleration.
> If you have any questions about other specific deployment environments, please feel free to let us know.
>
> **Finally, I hope that my response has addressed your concerns, and I kindly ask for a more favorable evaluation of our work. Thank you!**

---

> ### Author Response · Authors · 2024-11-29
> **A kind reminder**
>
> Dear Reviewer MFoH,
>
> Thanks again for your valuable time and insightful comments. As the deadline for the Author/Reviewer discussion is approaching, it would be nice of you to let us know whether our answers have solved your concerns so that we can better improve our work. We are happy to provide any additional clarifications that you may need.
>
> Best regards,
>
> Authors

---

> > ### Comment · Reviewer_MFoH · 2024-11-29
> >
> > Thank you for the detailed response and experiments. I think most of my concerns have been addressed, so I will raise my score.

---

> > > ### Author Response · Authors · 2024-11-30
> > > **Response to Reviewer MFoH**
> > >
> > > Thank you for your hard work and recognition of our paper. We're glad to hear that you're satisfied.

---

### Official Review · Reviewer_FSq8 · 2024-11-04

**Soundness:** 3
**Presentation:** 3
**Contribution:** 3
**Rating:** 6
**Confidence:** 4

**Summary:**

This paper propose an adaption method for fine-tuning a dense LLM resulting into a sparse LLM which is more powerful and faster. The proposed algorithm works on sparsity design and low rank allocation. Experiements are evaluated on LLaMA and OPT models to demonstrate the better model performance and faster inference speed while comparing with other method at the same sparsity rate.

**Strengths:**

1. reasonable motivation, studying the sparsification of LLM while applying LoRA.
2. clear problem formulation and related work introduction at each subproblem.
3. detailed and summarized pseudocode for connecting each step and explaining the overall algorithm.
4. strong and promising experimental results on LLMs regarding both model performance and speedup.

**Weaknesses:**

1. the algorithm consists of many heuristics and is lack of step by step derivation, e.g., Eq. 7 and Eq. 9
2. some experiment details are missing and unclear.

**Questions:**

1. section "RMI for Sparsity" with Eq. 2 and Eq. 3 are not used in the proposed algorithm which can confuse readers for their purpose. this paper could shorten this presentation.
2. Eq. 7 has a hidden assumption that higher rank setting can help reduce reconstruction loss. According to Eq. 1, under current problem formulation, higher rank does not change anything given the sparsity mask M. Thus, the design of Eq. 7 is not useful. Similarly, this question also applies to Eq. 9. Why Eq. 9 is performed at each step and what happens if Eq. 9 became too large?
3. It is unclear how rank increase and decrease is implemented in practice. For example, if current rank is 3 and you want to increase to 4, do you initialize the additional vector randomly? if current rank is 4 and you want to decrease to 3, do you perform rank reduction by singular values magnitude (if you use svd)?
4. It should be discussed the computation complexity. In particular, the sparsity mask computation can be slow. Given the algorithm is executed at each fine-tuning step, the overall computation time should be reported.
5. Experiment result table report "SparseGPT with LoSA", while the Algorithm 1 input is dense weight of LLM. Which part of dense weight of SparseGPT you work on?
6. Which layer of LLM and weight matrix (Q,K,V, etc) you run experiment with?
7. What is the outcome of SparseGPT with LoRA? Does it become a dense LLM? If so, what does sparsity 50% mean for SparseGPT with LoRA?
8. Is there any guideline for setting fine-tuning steps? In Table 1, T=5, why 5 steps can be sufficient for completing LoRA training?

---

> ### Author Response · Authors · 2024-11-21
> **Response to Reviewer FSq8 (1/2)**
>
> **Thanks for your careful review and comments! Hopefully the following contents could answer your questions.**
> > W1: the algorithm consists of many heuristics and is lack of step by step derivation, e.g., Eq. 7 and Eq. 9
>
> We thank the reviewer for pointing out this issue. In this paper, the determination of sparsity rates for LLMs is derived from the Information Bottleneck theory, rather than relying on heuristic designs. For the low-rank adaptation, we indeed employ heuristic approaches to derive Eq. 7 and Eq. 9. However, the rank of the matrices serves as a hyperparameter, making it challenging to determine the rank for LoRA through theoretical derivations. Many existing studies [1–3] aim to gradually adjust the rank during the fine-tuning process to achieve an optimal rank allocation for LoRA. This paper advocates for the dynamic allocation of the rank parameter budget during sparse LLMs finetuning process, based on the sparse reconstruction errors across different layers. Therefore, we designed Eq. 7 and Eq. 9. Furthermore, while our rank determination is based on heuristic formulas, the ablation studies in Table 5 validate the effectiveness of our adaptive rank adjustment for sparse LLMs. In the future, we will try to prove the validity of Eq. 7 and Eq. 9 with rigorous mathematical proofs.
>
> [1] AdaLoRA: Adaptive budget allocation for parameter-efficient fine-tuning.
>
> [2] Sparse low-rank adaptation of pre-trained language models.
>
> [3] ALoRA: Allocating Low-Rank Adaptation for Fine-tuning Large Language Models.
>
> > W2: some experiment details are missing and unclear.
>
> Thank you for your valuable feedback. Perhaps the experiment details you are talking about are mentioned in the question section? We will answer questions about the experiment details in the question section. If you feel that other experiment details are missing and unclear, please explain in more detail.
>
> > Q1：section "RMI for Sparsity" with Eq. 2 and Eq. 3 are not used in the proposed algorithm which can confuse readers for their purpose. this paper could shorten this presentation.
>
> Thank you for your suggestions. In the section "RMI for Sparsity," we derived the RMI metric to determine the importance of each layer in LLMs. While Eq. 2 and Eq. 3 are not directly used in the proposed algorithm, they ultimately lead to the derivation of Eq. 4 and Eq. 5. Eq. 4 and Eq. 5 provide the formulas for determining layer importance and calculating sparsity rates. Therefore, we included this section in the paper to present the complete derivation. We have streamlined the expressions in this section as much as possible. If you have any further questions or concerns regarding the presentation of this section, please let us know.
>
> > Q2: Eq. 7 has a hidden assumption that higher rank setting can help reduce reconstruction loss. According to Eq. 1, under current problem formulation, higher rank does not change anything given the sparsity mask M. Thus, the design of Eq. 7 is not useful. Similarly, this question also applies to Eq. 9. Why Eq. 9 is performed at each step and what happens if Eq. 9 became too large?
>
> 1. When the rank $r$ of matrices $\boldsymbol{B} \in \mathbb{R}^{c_{cout}\times r}$ and $\boldsymbol{A} \in \mathbb{R}^{r \times c_{in}}$ increases, the parameters of both $\boldsymbol{B}$ and $\boldsymbol{A}$ in Eq. 1 are increased. After fine-tuning, $\boldsymbol{B}$ and $\boldsymbol{A}$ can better compensate for the reconstruction error $\mathcal{L}$ between the sparse and dense model.
> 2. Additionally, the average rank of all layers is set to $\Omega$ in Eq. 7. Under this constraint, we allocate a larger rank to layers with higher reconstruction error, as these layers will require more fine-tuning parameters to reduce the error effectively.
> 3. Because we adopt a progressively increasing sparsity strategy, the sparsity rate gradually grows, which leads to an increase in the reconstruction error of the sparse model. To address this issue, we increase the average rank of all layers as the sparsity step increases in Eq. 9. Therefore, both Eq. 7 and 9 are valid. We investigate the impact of rank on the perplexity of the final model in Figure 2. Increasing the rank appropriately can effectively reduce perplexity, leading to a better recovery of the sparse model’s performance. However, since the fine-tuning samples are fixed, further increasing the rank results in insufficient training of low-rank adaptation, which causes an increase in perplexity.

---

> ### Author Response · Authors · 2024-11-21
> **Response to Reviewer FSq8 (2/2)**
>
> > Q3: It is unclear how rank increase and decrease is implemented in practice. For example, if current rank is 3 and you want to increase to 4, do you initialize the additional vector randomly? if current rank is 4 and you want to decrease to 3, do you perform rank reduction by singular values magnitude (if you use svd)?
>
> 1. For training stability, when two low-rank matrices $\boldsymbol{B} \in \mathbb{R}^{c_{out}\times r}, \boldsymbol{A} \in \mathbb{R}^{r \times c_{in}}$ increase their rank, we concatenate random Gaussian initialized parameters $\mathcal{N} (0, \sigma^2)$ to $\boldsymbol{A}$ and zeros to $\boldsymbol{B}$. The above initialization operation is the same as LoRA, so the layer’s output remains unchanged before and after new parameters added.
> 2. When two low-rank matrices $\boldsymbol{B}, \boldsymbol{A}$ decrease their rank, the converted $\boldsymbol{B}^{'}, \boldsymbol{A}^{'}$ directly inherit parameters from the original matrix $\boldsymbol{B}, \boldsymbol{A}$ and the extra parameters are discarded.
>
> > Q4: It should be discussed the computation complexity. In particular, the sparsity mask computation can be slow. Given the algorithm is executed at each fine-tuning step, the overall computation time should be reported.
>
> We have discussed the computation complexity in Section 3.6 and shown the overall computation time in Table 8. Specifically, since LoSA uses multiple sparsification operations and needs to obtain sparse masks multiple times, the overall computation time of LoSA is longer than that of LoRA. Nevertheless, LoSA provides better accuracy and lower inference latency compared to LoRA, and it only requires about an hour of fine-tuning, which we believe is a worthwhile trade-off.  If you have any questions about our analysis, please let us know.
>
> > Q5:Experiment result table report "SparseGPT with LoSA", while the Algorithm 1 input is dense weight of LLM. Which part of dense weight of SparseGPT you work on?
>
> In Algorithm 1, we start with a dense LLM weight as input, and SparseGPT determines the sparsity mask for the LLM, which is then applied to the dense weight to obtain a sparse weight. The LoSA method fine-tunes the sparse LLM obtained by SparseGPT to improve the accuracy of the sparse LLM.
>
> > Q6: Which layer of LLM and weight matrix (Q,K,V, etc) you run experiment with?
>
> We apply sparsity to all Linear layers in the Attention and MLP modules following the setup of the Wanda and SparseGPT methods. For the LLaMA model, this includes sparsifying the weight matrices of Q, K, V, Out Proj, Up Proj, Down Proj, and Gate Proj, while excluding the final LM Head. Additionally, we add low-rank adaptation to all sparse weight matrices. In Table 11, we present an ablation study that shows the effect of applying low-rank adaptation to only certain sparse weight matrices (e.g., Q, K or Q, K, V, O). The experimental results demonstrate that applying low-rank adaptation to all sparse weight matrices yields the best performance.
>
> > Q7:What is the outcome of SparseGPT with LoRA? Does it become a dense LLM? If so, what does sparsity 50% mean for SparseGPT with LoRA?
>
> Since the LLM weights are sparse and the LoRA weights are dense, merging the dense LoRA weights into the sparse LLM weights would cause the LLM weights to become dense as well. Therefore, for a sparse LLM, the LoRA branch can only be retained without merging, in order to preserve the compression and acceleration benefits of the sparse LLM. However, retaining the LoRA branch increases the inference time of the sparse LLM. In contrast, the LoSA weights are also sparse and can be merged into the sparse LLM weights, avoiding any increase in inference time for the sparse LLM. For a comparison of inference time between LoRA and LoSA, please refer to Table 9.
>
> > Q8: Is there any guideline for setting fine-tuning steps? In Table 1, T=5, why 5 steps can be sufficient for completing LoRA training?
>
> We have presented the ablation study of the fine-tuning steps T in Figure 2 and discussed the impact of different fine-tuning steps on the final perplexity in Section 3.5. Steps T determines the frequency at which the sparsity rate increases. A larger T means the sparsity rate increases more slowly, and fewer parameters are removed each time. For each fine-tuning step, we first apply sparsity to the LLM, and then fine-tune the sparse LLMs using 2K Alpaca-GPT4 samples. Since we use a sufficient number of fine-tuning samples, this is enough to train LoRA. We show the impact of different values of T on the final perplexity while keeping the number of fine-tuning samples constant. Increasing T appropriately can effectively reduce the model’s perplexity. However, a larger T may result in insufficient training of the model after each sparsification, which in turn leads to a further increase in perplexity.
>
> **Finally, I hope that my response has addressed your concerns, and I kindly ask for a more favorable evaluation of our work. Thank you!**

---

> ### Author Response · Authors · 2024-11-29
> **A kind reminder**
>
> Dear Reviewer FSq8,
>
> Thanks again for your valuable time and insightful comments. As the deadline for the Author/Reviewer discussion is approaching, it would be nice of you to let us know whether our answers have solved your concerns so that we can better improve our work. We are happy to provide any additional clarifications that you may need.
>
> Best regards,
>
> Authors

---

### Official Review · Reviewer_VhBK · 2024-11-05

**Soundness:** 3
**Presentation:** 2
**Contribution:** 2
**Rating:** 8
**Confidence:** 3

**Summary:**

The paper introduces LoSA (dynamic Low-rank Sparse Adaptation), a novel method for fine-tuning sparse Large Language Models (LLMs). LoSA addresses the performance degradation often observed in sparse LLMs by integrating low-rank adaptation (specifically inspired by LoRA) directly into the sparsity process.  It does so by dynamically sparsifying the low-rank adaptation outcomes to ensure compatibility with the sparse LLM weights and enable merging post-training.  Furthermore, LoSA uses Representation Mutual Information (RMI) to dynamically adjust layer-wise sparsity rates and allocates the rank of low-rank adaptation based on layer-wise reconstruction errors.  Experiments on various LLMs (LLaMA variants, OPT, Vicuna) demonstrate that LoSA significantly improves perplexity and zero-shot accuracy compared to existing sparsity methods and LoRA, while maintaining inference speedup.

**Strengths:**

* Addresses a relevant problem: Performance degradation in sparse LLMs is a known issue, and LoSA offers a practical solution.

* Novelty: Integrating sparsification into the low-rank adaptation process and dynamically adjusting sparsity/rank based on RMI and reconstruction errors are novel ideas.

* Strong empirical results: The experimental results show consistent improvements across various LLMs and sparsity levels.

* Inference efficiency: LoSA preserves the inference speed advantages of sparsity by merging the adapted weights.

**Weaknesses:**

na

**Questions:**

1. Can you comment (qualitiatively) on the applicability of the mutual information based layer-sensitivity method to other compression techniques -- e.g., would it work for quantization, or if we were jointly quantizing and sparsifying?

2. You have a brief section on N:M sparsity, where you fix M=8. How easy is it to extend this to also determine the right value of M for different layers? Relatedly, any notes on smaller values of M (like M=4, found in Nvidia GPUs as 2:4 sparsity)?

3. Can you add more color to the cost of your proposed method during training (i.e., impact on training time). There is an annotation that it takes 48 seconds for LLama-2-7B. Can you clarify: (i) what is the relative contribution to the overall step time, (ii) if the computations are performed in every step (or less frequently like every k steps -- or if it is possible to do it less frequently).

---

> ### Author Response · Authors · 2024-11-21
> **Response to Reviewer VhBK**
>
> **We greatly appreciate your time, detailed comments, and valuable suggestions. Below, I will address each of your questions one by one.**
>
> > Q1：Can you comment (qualitiatively) on the applicability of the mutual information based layer-sensitivity method to other compression techniques -- e.g., would it work for quantization, or if we were jointly quantizing and sparsifying?
>
> 1. We determine the importance of each layer based on the mutual information metric, assigning corresponding sparsity rates to each layer. Layers deemed more important are allocated lower sparsity rates. Similarly, we can also determine the importance of each layer using the mutual information metric and assign an appropriate number of quantization bits to each layer for quantization, with more important layers allocated higher bit precision.
> 2. Likewise, we can use the mutual information metric to determine the compression rate for each layer in a jointly quantized and sparse model. Specifically, based on the layer importance derived from the mutual information metric, more important layers are assigned lower compression rates. Based on the obtained compression rate, we can further determine the sparsity rate and quantization bits of each layer.
>
> > Q2: You have a brief section on N:M sparsity, where you fix M=8. How easy is it to extend this to also determine the right value of M for different layers? Relatedly, any notes on smaller values of M (like M=4, found in Nvidia GPUs as 2:4 sparsity)?
>
> 1. We extend LoSA to N:M sparsity and adopt a mixed N:8 sparsity (N refers to non-zero weights) configuration, and M value is fixed for different layers, you can set it to any value (such as 8 or 4). Meanwhile, we allow different layers to have distinct N values while maintaining a constant overall sparsity ratio. We assign lower N values to less important layers and the N value for each layer are determined using the mutual information based sparsity method. Since mixed 2:8 sparsity is a specific type of unstructured sparsity pattern, it can also leverage the nm-vllm sparse inference engine to achieve accelerated inference on GPUs. The acceleration effect is as follows:
>
> |Speed|Dense|Mixed 2:8|
> |-|-|-|
> |Throughput (tokens/s)|57.35|133.40|
> |Speedup|1.00$\times$|2.33$\times$|
>
> 2. Secondly, we can also easily use a smaller value of M to obtain a mixed 2:4 sparsity model. We conducted experiments of mixed 2:4 sparsity on LLaMA2-7B obtained by Wanda method. The experimental results are as follows:
>
> |Method|Sparsity|WikiText2 PPL|HellaSwag|WinoGrande|BoolQ|OBQA|PIQA|ARC-e|ARC-c|Average|
> |-|-|-|-|-|-|-|-|-|-|-|
> |Dense|0%|5.12|57.17|68.90|77.74|31.40|78.07|76.39|43.52|61.88|
> |Wanda|2:4 sparsity|11.02|40.92|62.43|67.65|24.20|70.84|61.78|31.20|51.29|
> |w. LoRA|2:4 sparsity|8.27|50.37|64.80|72.81|27.60|75.19|69.40|35.58|56.54|
> |**w. LoSA**|**Mixed 2:4 sparsity**|**7.72**|**51.85**|**66.01**|**74.51**|**29.70**|**76.54**|**71.08**|**37.26**|**58.14**|
>
> LoSA improves accuracy for mixed 2:4 sparsity and outperforms LoRA. We also present experimental data showing the use of the nm-vllm sparse inference engine to accelerate inference on GPUs for LLaMA2-7B with mixed 2:4 sparsity.
>
> |Speed|Dense|Mixed 2:4|
> |-|-|-|
> |Throughput (tokens/s)|57.35|98.35|
> |Speedup|1.00$\times$|1.71$\times$|
>
> > Q3: Can you add more color to the cost of your proposed method during training (i.e., impact on training time). There is an annotation that it takes 48 seconds for LLama-2-7B. Can you clarify: (i) what is the relative contribution to the overall step time, (ii) if the computations are performed in every step (or less frequently like every k steps -- or if it is possible to do it less frequently).
>
> 1. The layer-wise sparsity rates for the LLaMA-2-7B can be determined in an end-to-end time of only 48 seconds on a single NVIDIA A100 80GB GPU using our method. Moreover, since we adopt a progressive sparsification approach, we use a frequency T = 5, where the sparsity rate increases from 0 to the target sparsity rate. Therefore, determining the layer-wise sparsity rates takes a total of 48 seconds $\times$ 5 = 4 minutes. However, the total fine-tuning time for the LLaMA-2-7B is 45 minutes, making the time spent on sparsity rate computation relatively few compared to the overall fine-tuning time.
> 2. Of course, we can adjust the sparsification frequency T, either increasing or decreasing it, which will proportionally increase or decrease the time required for sparsity rate computation. Additionally, in Figure 2, we investigate the impact of different sparsification frequency T on the final perplexity of the model. Increasing T appropriately can effectively reduce the model’s perplexity. However, a larger T may result in insufficient training of the model after each sparsification, which in turn leads to a further increase in perplexity.
>
> **Finally, I hope that my response has addressed your concerns. Thank you!**

---

> ### Comment · Reviewer_VhBK · 2024-11-26
>
> Thank you for your responses. It addresses most of my questions.
>
> I think your response to [Q3] can use a little more details. I understand the overall (end-to-end) overhead of your method on training time. Would it be possible to annotate the specifics I requested: (i) what is the overhead of performing your method relative to an _individual_ step, (ii) how frequently is it performed?
>
> For example, if your training run is: 100k steps @ 0.05 ms per step := 5000 seconds (total) + your sparsity approach is applied every 20k steps and takes 10 seconds when applied := 50 seconds (total). Thus total runtime is 5050 seconds, of which 50 seconds is spent in your sparsity approach. Then:
> (i): Sparsity approach takes 50 seconds vs. 0.05 ms per step.
> (ii): The sparsification is done 5 times across the entire training duration.
>
> (A): I am inferring the 5 times in (ii) is the same as T=5 in your terminology. Correct?
> (B): I am inferring the 50 seconds in (i) is the same as 48 seconds in your terminology. Correct?
> (C): If you can annotate either: the time per step (the equivalent of 0.05 ms), or total steps (100k), the reader can derive the remainder.
>
> I am interested in (i) simply to understand what is the relative overhead, and because for a practitioner it can dictate how they may be willing to set the frequency in (ii). Or alternatively, it can pinpoint a computational bottleneck that the systems community can work on alleviating (as future work). As a different example, second-order optimizers are promising but expensive, so we either run them infrequently, or improve systems support for the relevant operations. I think it would be helpful for readers to understand the same calculus for your approach.
>
> Thank you again for your explanations and good work. My decision was originally Accept, and I will retain this score.

---

> > ### Author Response · Authors · 2024-11-27
> > **Response to Reviewer VhBK**
> >
> > Thank you for your further response and recognition of our work. I appreciate the opportunity to provide more detailed annotations and clarifications. Our further response to [Q3] is as follows:
> >
> > 1. Our LoSA method includes the calculation of LLMs sparsity rate, determination of sparse mask and determination of LoRA rank. The determination of sparse rate takes 48 seconds, the determination of sparse mask (using Wanda method) takes 66 seconds, and the determination of LoRA rank takes 46 seconds, so the total time of our LoSA method (sparsity rate + mask + rank) is 48+66+46=160 seconds. After completing above, we start training the weights of LoRA. The total number of training samples is 10K, and the batch size is 4, so the total training is 2500 steps, and each step takes 0.76 seconds.
> >
> > 2. Therefore, our training run is: 2500 steps @ 0.76 s per step := 31.7 minutes (total) + our LoSA approach is applied every 500 steps and takes 160 seconds when applied := 13.3 minutes (total). Thus total runtime is about 45 minutes, of which 13.3 minutes is spent in our LoSA approach. Then: (i): LoSA approach takes 160 vs. 0.76 seconds per step. (ii): The LoSA approach is done T=5 times across the entire training duration.
> >
> > 3. Overall, the computational overhead of our method is still very low and we can effectively fine-tune sparse LLMs in hours. We agree that understanding and addressing computational bottlenecks is critical. We plan to further optimize the computational efficiency of our method in future work, potentially collaborating with the systems community to improve hardware support for sparsity-driven training.
> >
> > Thank you again for your support of our work.

---

### Author Response · Authors · 2024-11-21
**General Response to All Reviewers and ACs**

We extend our heartfelt thanks to all reviewers and area chairs for their time and efforts in reviewing our paper and providing insightful comments. We are pleased that the reviewers have recognized our contributions, which we summarize below:
1. **Novelty.** We propose a novel and efficient fine-tuning method named LoSA for sparse LLMs, which seamlessly integrates low-rank adaptation into LLM sparsity within a unified framework and dynamically adjusting sparsity/rank based on mutual information and reconstruction errors. (Reviewer VhBK and MFoH)
2. **Strong and comprehensive empirical results.** The experimental results show consistent improvements across various LLMs and sparsity levels. Our paper’s experiments cover multiple architectures (LLaMA, Vicuna, OPT) and sizes, presenting results across different sparsity ratios. Performance improvements in both perplexity and zero-shot accuracy underscore LoSA’s adaptability across models and size. (All reviewers)
3. **Inference efficiency.** Our LoSA method preserves the inference speed advantages of sparsity by merging the low-rank adaptation weights. LoSA achieves significant inference speedups on CPUs and GPUs, which is crucial for deploying LLMs in resource-constrained environments. (Reviewer VhBK, FSq8, MFoH and VoNN)

We also appreciate the constructive suggestions and concerns raised by the reviewers, which have led to productive discussions and additional experiments. We have made appropriate revisions, with all changes marked in blue for easy reference. We believe these revisions further strengthen our paper. We summarize our major changes as follows:

1. Make minor corrections to the errors in Eq.1 and Eq.5. (Section 2.1 and 2.2)
2. Add implementation details of parameter initialization when increasing or decreasing the rank of low-rank adaptations. (Section 2.3)
3. Add comparisons and discussions with previous similar works. (Section B)
4. Add comparison results of LoSA and Sparse LoRA. (Section C.5)
5. Add ablation experiments on the OPT model to verify the effectiveness of RMI-based sparsity and reconstruction-based rank allocation. (Section C.6)
6. Extend our LoSA method to structured pruning and compare with LLM-Pruner, SliceGPT, LoraPrune, and LoRAShear baselines. (Section D)
7. Add experimental results of LoSA fine-tuning some latest LLM post-training sparsity works. (Section E)
8. Add experimental comparison with SoRA and AdaLoRA baselines. (Section F)
9. Add discussion on mixed N:M sparsity setting experimental results. (Section G)
10. Add computational cost statistics of RMI-based sparsity and reconstruction-based rank allocation for models with different parameter sizes. (Section H)
11. Add analysis of layer-wise sparsity rate obtained by RMI. (Section I)

Finally, **we have addressed all the raised questions and concerns**, and we present detailed responses to individual reviewers below. We look forward to any additional comments.

---

### Author Response · Authors · 2024-11-25
**A kind reminder regarding our response**

Dear Reviewers,

We have put in many efforts to address the raised questions and concerns. As the ICLR rebuttal period is approaching its end, we kindly remind you to review our submitted response. Your feedback is essential for finalizing our work.

Thank you for your attention.

Best regards,

The Authors

---

### Meta-Review · Area_Chair_oy6q · 2024-12-20

**Metareview:**

This paper introduces "Dynamic Low-Rank Sparse Adaptation" (LoSA), a novel framework designed to enhance the performance of sparse LMMs by integrating low-rank adaptation within their sparsity structures. LoSA dynamically adjusts layer-wise sparsity rates and low-rank allocations based on Representation Mutual Information (RMI) and layer-specific reconstruction errors. This approach ensures efficient compatibility between sparse and low-rank structures, maintaining inference speed while significantly improving model accuracy and perplexity across various LLM architectures like LLaMA, OPT, and Vicuna, without increasing inference latency.

**Strengths.**

All reviewers found the paper well-motivated, clearly written, and straightforward to follow. Also, they admired the extensive and promising experimental results across LLM architectures like LLaMA, Vicuna, and OPT, which demonstrate significant performance improvements and inference speedups, showing potential for deployment in resource-constrained environments.

**Weaknesses**

The reviewers highlighted key weaknesses in the submission, such as the absence of comparisons with recent pruning studies (raised by most reviewers), limited novelty, inadequate computational analysis of the RMI method, and insufficient implementation details. Additionally, they noted marginal improvements in several experimental results. Some minor issues, including the unconventional definition of sparsity rate, also caused confusion.

**Conclusion**

In my view, the authors have effectively addressed most of the previously identified weaknesses and have made significant enhancements to the experimental section of their paper. The proposed method, while relatively simple, has merit when compared to recent baselines. Regrettably, despite repeated reminders from the Area Chair, several reviewers have not acknowledged the authors' rebuttal efforts. Notably, the review by Reviewer 7hsM, which received a rating of 3 and a confidence level of 4, was less substantial and focused solely on comparisons with recent baselines, a concern that the authors have adequately addressed in their rebuttal. Furthermore, Reviewer VoNN's primary critique is that different components of the paper have appeared in previous work; hence, the paper lacks novelty. However, the authors have appropriately credited prior work in the revised paper, particularly regarding components like the RMI. I agree with the authors that integrating these methods into their proposed framework does introduce a novel element, even if it's not groundbreaking. Considering these factors, I believe the paper is a step in the right direction. I join the majority of the reviewers in positively evaluating this work and believe that it will contribute value to our esteemed conference. Consequently, I vote for accepting the paper.

**Additional Comments On Reviewer Discussion:**

This paper elicited a highly polarized set of reviews. Despite my efforts to engage the reviewers in a discussion during the review period to reach a consensus on the paper’s merits and shortcomings, they did not participate in any discussion. This made the evaluation of this paper very challenging.

After thoroughly reading the paper and considering the reviewers' feedback, I observed that the majority of the concerns raised by the reviewers have been properly addressed in the authors' rebuttal. Regrettably, several reviewers did not respond further and failed to acknowledge the authors' rebuttal efforts. Based on my evaluation, I would rate the updated paper as a solid 6 (or even 7, if that were an option). Given the significance and timeliness of the topic, I believe the work represents a positive advancement. Therefore, I vote to accept the paper.

---

### Decision · Program_Chairs · 2025-01-22

Accept (Poster)